# Role of Glycans on Key Cell Surface Receptors That Regulate Cell Proliferation and Cell Death

**DOI:** 10.3390/cells10051252

**Published:** 2021-05-19

**Authors:** Yin Gao, Xue Luan, Jacob Melamed, Inka Brockhausen

**Affiliations:** 1Key Laboratory for Molecular Enzymology and Engineering of Ministry of Education, School of Life Sciences, Jilin University, Changchun 130012, China; yingao@jlu.edu.cn (Y.G.); luanxue19@mails.jlu.edu.cn (X.L.); 2Department of Biomedical and Molecular Sciences, Queen’s University, Kingston, ON K7L3N6, Canada; jacob.melamed@queensu.ca

**Keywords:** *N*-glycans, *O*-glycans, receptors, signaling, tyrosine kinase, TGFR

## Abstract

Cells undergo proliferation and apoptosis, migration and differentiation via a number of cell surface receptors, most of which are heavily glycosylated. This review discusses receptor glycosylation and the known roles of glycans on the functions of receptors expressed in diverse cell types. We included growth factor receptors that have an intracellular tyrosine kinase domain, growth factor receptors that have a serine/threonine kinase domain, and cell-death-inducing receptors. *N*- and *O*-glycans have a wide range of functions including roles in receptor conformation, ligand binding, oligomerization, and activation of signaling cascades. A better understanding of these functions will enable control of cell survival and cell death in diseases such as cancer and in immune responses.

## 1. Introduction

The plasma membrane of mammalian cells has important functions in the activation of signaling pathways and regulation of cell proliferation and cell death [1,2,3]. Cell membrane-bound receptors responsible for these biological functions are type I single spanning transmembrane proteins that are usually heavily glycosylated. Many of the ligands for these receptors are also glycoproteins that, upon binding, induce oligomerization or clustering of receptors that initiates signal transduction. Mechanisms to reduce signaling include internalization of receptor complexes to endosomal compartments. The crucial first step of receptor function is the interaction between ligand and receptor (or co-receptor) in the extracellular domain. Receptor-bound glycans protrude from the core of the protein into the extracellular space. The multiple *N*-glycan chains in this domain have the potential to control the conformation and accessibility of specific ligand binding sites. The second event is the clustering of receptor molecules where protein–protein interactions can also be influenced by the presence of glycans. The subsequent events include activation of protein kinase activities or recruitment of intracellular ligands leading to phosphorylation of the cytoplasmic domains. These processes trigger several specific signaling pathways that ultimately regulate gene transcription and protein expression [2,4].

In addition to specific ligands that bind to receptors, other glycoconjugates also influence receptor function, including ligand analogs (agonists) and inhibitors. For example, heparin has been shown to bind several receptors thus modifying their affinity and functions [5]. Components of the cell membrane such as lipids and glycosphingolipids in lipid rafts are also involved in regulating receptor functions.

There is a wealth of cell surface receptors with broad tissue distribution. These receptors initiate signal transduction utilizing shared intracellular proteases, kinases and transcription factors that regulate cellular survival, cell death, migration and many other biological phenomena. Many receptors, as well as the glycosyltransferases (GTs) that glycosylate proteins, are expressed in a cell-type-specific fashion, thus producing a heterogeneous array of cell-specific glycan structures and attachment sites. These cell-type-specific glycans may have varying regulatory effects on receptor activity (Figure 1).

Receptors often exist as variants that slightly differ in amino acid sequence and ligand affinity. These isoforms, together with naturally occurring mutants and mutagenesis studies, help to identify receptor functions at the molecular level [6]. Some of these mutations are associated with pathology suggesting the essential roles of certain receptors [7,8]. A number of receptor variants are stabilized in the membrane via glycosylphosphatidyl-inositol- (GPI-) anchors. In order to initiate signaling, they need to associate with signaling receptors. Secreted isoforms of receptors can block receptor functions as decoy receptors [4,9].

Protein-bound glycans have been shown to be essential for protein folding and stability, intracellular transport, epitope expression and cell surface interactions with carbohydrate binding proteins of the immune system. There are also indirect effects of glycans, mediated by carbohydrate binding proteins such as galectins. 

Despite the abundance of receptor-bound glycans, their exact structures and roles have received very little attention. *N*-glycosylation sites are easily detectable as Asn-X-Ser/Thr-Y (where X, Y is not Pro), but it is difficult to determine the site-specific glycan structures. *O*-glycosylation sites are not as easily predicted, and we do not know much about receptor *O*-glycosylation. Lectin and antibody binding to purified receptors and mass spectrometry have been used as tools to determine the structures and attachment sites of glycan epitopes. 

The roles of glycans have mainly been studied by mutagenesis of *N*-glycosylation sites, enzymatic removal of glycans and by treating cells with glycosylation inhibitors. For example, tunicamycin blocks the formation of the *N*-acetylglucosamine (GlcNAc)-diphosphate-dolichol intermediate for *N*-glycan biosynthesis. However, tunicamycin treatment also induces endoplasmic reticulum (ER) stress which may be linked to apoptosis [10]. Similarly, inhibition of *O*-glycan extension by *N*-acetylgalactosamine (GalNAc)α-benzyl can induce stress through intracellular accumulation of intermediates [11]. Another approach to studying the relationship between glycosylation and receptor signaling is knockdown or overexpression of specific GT genes [12]. These experiments reveal the involvement of glycans in receptor and ligand expression, secretion and intracellular transport [13,14]. However, this information is scarce, and many studies focus on the structure and function of the cytoplasmic domain of receptors (oncoproteins) and on the development of signaling inhibitors. The cytoplasmic domain of receptors may be glycosylated by *O*-GlcNAc that plays a role in protein phosphorylation [15]. 

This review summarizes the current knowledge of the importance and roles of glycans in ligand binding and in receptor functions, in regulating proliferation and regeneration or cell death in somatic and neuronal cells. The tyrosine kinase growth factor receptors that will be examined include epidermal growth factor receptor (EGFR), hepatocyte growth factor receptor (MET), fibroblast growth factor receptor (FGFR), vascular endothelial growth factor receptor (VEGFR), insulin receptor (INSR) and insulin-like growth factor receptor (IGFR). Among receptors for neurotrophic factors, RET is a dimeric receptor that requires GPI-anchored co-receptors for binding glial cell-derived neurotrophic factor (GDNF). Tyr kinases TrkA, TrkB and TrkC all participate in regulating proliferation and neurite outgrowth in neurons upon binding nerve growth factor (NGF), GDNF or brain-derived nerve growth factor (BDNF) [16,17,18]. The p75 receptor also binds NGF but is related to the tumor necrosis factorα (TNFα) receptor family and can induce apoptosis as well as survival. Additional glycosylated receptors include Ser/Thr kinases, e.g., transforming growth factor receptor (TGFR) [16,19,20,21]. We also discuss the glycosylation of trimeric death receptors Fas and receptors for TNFα and TNFα-related apoptosis-inducing ligand (TRAIL) that regulate both cell death and survival. We highlight the many roles of glycans that surround the extracellular domains of these receptors.

## 2. Tyrosine Kinase Receptors

Tyrosine kinase (Trk) receptors are type I membrane proteins found on plasma membranes that form dimers upon ligand binding, thus triggering signaling cascades (Table 1). The extracellular domain (ectodomain) is heavily *N*-glycosylated and contains the ligand binding site (Figure 2A–C). The cytoplasmic domain contains the Tyr-kinase that is activated to self-phosphorylate specific Tyr residues and bind to intracellular proteins resulting in activation of pathways of cell survival, growth or angiogenesis. Growth factors are ligands for cell-type-specific Trk receptors with other proteins participating.

### 2.1. Epidermal Growth Factor Receptor, EGFR

Epidermal growth factor receptor (EGFR, HER1) belongs to the ErbB family of receptor tyrosine kinases, including ErbB-2 (HER2), ErbB-3 (HER3) and ErbB-4 (HER4), that transduce cell signaling upon ligand binding [1,22]. EGFRs are important in epithelial and mesothelial cells as drivers of cell proliferation, migration, angiogenesis, differentiation and epithelial repair. The expression of EGFR has been found to be upregulated in cancer and EGFR is a major target for anticancer drug discovery as it inhibits apoptosis and promotes proliferation of cancer cells. EGFR is a type I transmembrane protein whose extracellular domain consists of four subdomains, with domains I and III being involved in binding EGF family members, and domain II and IV participating in receptor dimerization [23]. Without EGF stimulation, subdomains II and IV of EGFR maintain a tethered conformation preventing receptor dimerization and autophosphorylation [24]. Binding of EGF opens the tethered conformation, triggering dimerization and receptor activation through initiation of Tyr kinase activity [25,26,27]. Tyr kinase can also phosphorylate other proteins, e.g., Tyr in a YEKV motif of the cytoplasmic tail of MUC1 which allows binding of MUC1 to c-Src and β-catenin [28]. Adapter proteins participate in the cytoplasmic phosphorylation and signaling process. Downstream phosphorylation events involve mitogen-activated protein Ser/Thr kinases (MAPK) and extracellular signal-regulated kinases (ERK).

The EGFRs are all heavily glycosylated, with 7 to 13 *N*-glycosylation sites in the extracellular domain (Figure 2A). Thus, *N*-glycans cover a significant amount of space surrounding the proteins. *N*-glycans have been shown to play a role in conformational stability and clustering of receptors, in ligand binding and receptor activation [29]. Studies of *N*-glycosylation mutants of ErbB3 showed that receptor dimerization and activation were facilitated by the absence of *N*-glycans [30,31]. Increased cell growth was demonstrated in CHO cells and in tumors in athymic mice. However, mutation of *N*-glycosylation sites in domain III of ErbB3 reduced its cell surface expression. Removal of Asn418 was particularly important in enhancing cell proliferation, likely because glycosylation at this Asn residue has an important role in receptor conformation and the ability to associate with other receptor molecules (Table 2).

The extracellular region of EGFR contains 13 *N*-glycosylation sites but mass spectrometry of the receptor expressed in human epidermoid carcinoma A431 cells showed that only 11 sites were occupied by *N*-glycans [58]. The EGFR expressed in CHO cells carried oligomannose and complex chains [59]. An interesting finding was that the extracellular domain of EGFR expressed in CHO cells had only a fraction of the *N*-glycosylation sites occupied. One of these sites had an *N*-glycan attached to the first Asn of the Asn-Asn-X-Cys sequence where a Cys residue replaced the function of Ser/Thr that is normally found in *N*-glycosylation sites [60].

The activity of EGFR has been reported to be glycosylation-dependent [61]. The glycosylation of EGRF also affects resistance to tyrosine kinase inhibitors (TKI) [41,56,57], which is an important consideration for cancer treatment. The glycoforms of EGFR in cancer cells have been intensively investigated in recent years [31]. *N*-glycosylation of EGFR has been found to support the formation of noncovalent interactions between glycans and peptides of the extracellular domain of EGFR close to the ligand binding site. Molecular dynamics simulations suggested that *N*-glycans stabilize the EGFR binding site and provide strong interactions of EGFR with its ligand, with monoclonal antibodies, as well as with other receptor molecules [62].

Native EGFR undergoes autophosphorylation in response to EGF stimulation. Mutations at *N*-glycosylation sites Asn420 and Asn579 of EGFR weakens the tethered conformation of the receptor, thereby promoting receptor dimerization in absence of ligand induction, resulting in EGF-independent phosphorylation. Mutation at Asn579 also led to increased affinity of receptor to ligand and dimerization compared to native EGFR [32,33]. This shows the critical role of *N*-glycans in receptor function.

GlcNAc-transferase V (Gn-T V) adds the GlcNAcβ1-6 branch to complex *N*-glycans and is overexpressed in several types of cancers [63]. Knockdown Gn-T V showed no effect on EGF binding to EGFR, but resulted in reduced EGF-promoted activation of focal adhesion kinase and attenuation of the invasive phenotype of breast carcinoma cells [37]. In contrast, the expression of GlcNAc-transferase III (Gn-T III) that introduces the bisecting GlcNAc residue in hybrid-type or complex *N*-glycans is often downregulated in cancers and associated with cancer suppression. The bisected *N*-glycans appear to be enriched in specific areas of the brain, including the brainstem, and may play a role in neuronal cell differentiation [36]. Overexpression of Gn-T III in a human glioma cell line caused reduced EGF binding and EGFR phosphorylation, thus preventing EGFR activation. However, proliferation of glioma cells was stimulated by an unknown mechanism [35]. In rat pheochromocytoma cells, stimulation of both EGFR and integrin caused neurite outgrowth. Expression of Gn-T III reduced neurite outgrowth and EGFR activation by EGF binding and EGFR phosphorylation through the Ras/MAPK pathway. When expressed in HeLa cells that overexpressed Gn-T III, the ability of the receptor to bind to its ligand was reduced while the internalization of receptor complex and the EGF-induced ERK phosphorylation was increased [34].

A number of tumors have been shown to express high levels of α1,6-Fuc-transferase FUT8 that transfers Fuc to the core of complex *N*-glycans and plays a functional role in EGFR-mediated intracellular signaling [38,64,65]. Embryonic fibroblasts from FUT8^(-/-)^ mice exhibited strong inhibition of EGF-induced EGFR phosphorylation and EGFR-mediated JNK/ERK activation, compared to cells from FUT8^(+/+)^ mice. Complementation of FUT8 reversed the inhibitory effect and restored EGFR phosphorylation. This suggests that the *N*-glycan core Fuc residue is required for the interaction between EGF and EGFR, although it did not affect cell membrane expression of EGFR [38]. A similar role of *N*-glycan core fucosylation of EGFR was observed in human HEK293 and A549 cells, where increased fucosylation enhanced EGF-mediated dimerization and cellular signaling [39,47].

Lewis epitopes are Fuc-containing glycans found on *N*-glycans, *O*-glycans and glycoplipids and are synthesized by a family of α1,3-, α1,3/4- and α1,2-Fuc-transferases. EGFR is modified by Lewis epitopes, and especially Lewis^y^ [Fucα1-2Galβ1-4 (Fucα1-3) GlcNAc-] [66]. High levels of Lewis^y^ structures correlate with a poor prognosis in oral and other cancers. Knockdown of α1,2-Fuc-transferase FUT1 in oral squamous carcinoma cells prevented Lewis^y^ synthesis and was shown to promote cell migration through EGFR-mediated AKT and ERK activation pathways [40].

Another Fuc-transferase involved in the synthesis of Lewis^y^ is α1,3-Fuc-transferase FUT4, which transfers Fuc to GlcNAc in α1-3 linkage. Knockdown of either FUT1 or FUT4 in epidermoid carcinoma cells A431 reduced Lewis^y^ expression, showing that these two enzymes are major contributors to Lewis^y^ synthesis. Treatment of A431 cells with siRNA to decrease the expression of either FUT1 or FUT4 inhibited cell proliferation and blocked EGF-stimulated EGFR phosphorylation and MAPK. These siRNA-treated A431 cells showed significantly reduced tumor growth in mice [44].

In gastric cancer cells NCI-N87, a knockdown of FUT1 expression via shRNA led to reduced expression of Lewis^y^ on EGFR. EGFR degradation was enhanced, and EGF-induced cell migration was inhibited, while cell proliferation was not affected. [42]. Suppressing FUT1 expression in gastric, breast and lung cancer cells reduced the expression of ErbB2, inhibited ErbB2 phosphorylation and EGF-induced ERK1/2 activation. Transfection of FUT1 into ovarian cancer cells RMG-I to increase the expression of Lewis^y^ caused enhanced phosphorylation of EGFR. Consequently, cancer cell proliferation was promoted through the EGFR/PI3K signaling pathway [42].

One of the Fuc-transferases that synthesize sialyl-Lewis^x^ is FUT4. Knockdown of FUT4 in melanoma cells also resulted in decreased Lewis^y^ expression and reduced EGFR phosphorylation, and inhibited melanoma cell proliferation through EGFR-mediated MAPK signaling pathway. In addition, tumor growth in mice induced by these melanoma cells was reduced [45]. Lung cancer CL1 and A549 cells stably transfected with α1,3-Fuc-transferases FUT4 or FUT6 showed reduced EGFR dimerization and phosphorylation upon EGF induction. However, *N*-glycan core Fuc from EGFR increased EGFR dimer formation and cell growth [47]. These studies show that fucosylated epitopes are critical for the regulation of cell proliferation.

EGFR is also modified by sialyl-Lewis^x^ epitopes and FUT4 participates in the synthesis of the sialyl-Lewis^x^ structure that lacks the Fucα1-2 linkage [46]. The presence of sialic acid residues also appears to control receptor function. An increase of total sialylation of EGFR has been shown to reduce invasive properties of lung cancer cells A549, while sialidase treatment led to enhanced EGFR-mediated invasion and EGFR phosphorylation [47]. Moreover, the TKI-resistant cells showed increased sensitivity to gefitinib after sialidase treatment [48]. In HeLa cells, EGFR is associated with sialidase Neu3 in the plasma membrane, and overexpression of Neu3 promoted EGFR activation without increasing receptor expression [49]. Cells transfected with sialidase *Neu3* gene showed a reduction in sialylα2-6 linkages of EGFR and enhanced EGFR activation through Tyr kinase phospho-rylation without affecting mRNA and protein levels of EGFR. However, an inactive mutant form of Neu3 did not show this effect [49,50]. This suggests that both sialylation and fucosylation suppressed EGFR activation. Since Neu3 uses glycolipids as a preferred substrate, e.g., it converts GD1a to GM1, it is possible that EGFR is also regulated by the levels of glycosphingolipids in membranes or membrane lipid rafts.

Although these studies suggest a controlling function of receptor sialylation, other findings suggest the opposite. Sialidase-treated A549 cells were less sensitive to EGF-stimulated cell proliferation than A549 cells without sialidase treatment, and the TKI erlotinib showed no significant effects upon sialidase treatment [51]. Britain et al. [52] found that high expression levels of α2,6-sialyltransferase ST6Gal I that acts on complex *N*-glycans of EGFR correlates with EGF-triggered EGFR activation in pancreatic cancer cells [52,53]. The high expression of ST6Gal I also resulted in gefitinib resistance. The knockdown of ST6Gal I in ovarian cancer cells reduced EGFR activation and increased sensitivity to gefitinib-induced cell death. While α2,6-sialylation supported adhesion and migration of colon cancer cells and metastatic spread, ST6Gal I knockdown in colon cancer cells SW480 was shown to lead to EGF-induced phosphorylation of EGFR and ERK activation [54,67]. (Table 2).

EGFR is also a substrate for polypeptide GalNAc-transferases (GALNTs) that catalyze the first step of *O*-glycan biosynthesis [55] and transfer GalNAc residues to Ser/Thr to form the Tn antigen. Knockdown of GALNT2 in gastric cancer cells resulted in decreased Tn antigen and increased phosphorylation of EGFR.

Thus, GALNT2 may suppress the malignant phenotype of cancer cells by preventing the activation of EGFR and its downstream signaling pathway.

GALNT2 expression is reduced in hepatocellular carcinoma. These lower levels are in part responsible for cell proliferation through EGFR. After transfection of the *GALNT2* gene into hepatocellular carcinoma cells, EGF-induced cell proliferation, migration and invasion were reduced [56]. This indicated that *O*-glycosylation via GALNT2 suppressed the malignant phenotypes in liver cancer cells.

In contrast, GALNT2 is related to malignant phenotypes in a number of other cancer types. GALNT2 knockdown and overexpression in glioma cells demonstrated that GALNT2 was related to cell proliferation, migration and invasion through the EGFR/PI3K/AKT/mTOR signaling pathway [57]. Tumors grown in nude mice were less aggressive when GALNT2 expression was reduced in glioma cells. Although it remains to be directly shown that the Tn antigen of EGFR is responsible for the change in EGFR function, it has been suggested that GALNT2 may be a marker for malignant gliomas.

GALNT2 but not GALNT1 and GALNT3 is frequently overexpressed in oral squamous carcinoma. GALNT2 overexpression in oral squamous carcinoma cells was associated with increased migration and invasive phenotype through EGFR-mediated protein kinase B (AKT) phosphorylation and activation [41]. GALNT2 knockdown decreased phosphorylation of EGFR as well as the invasive phenotype.

The discrepancies found in these studies show that the roles of specific glycans of the EGFR may differ among cancer cell types, exhibiting different controls of receptor functions. Furthermore, the knockdown and overexpression of GTs in cells would also affect the glycosylation of other glycoproteins that may cooperate or compete with biological functions of EGFR.

### 2.2. Hepatocyte Growth Factor Receptor MET

Another cell surface-bound Tyr-kinase receptor is MET (Figure 2B), a receptor for hepatocyte growth factor (HGF) which regulates cell proliferation, differentiation, survival and morphogenesis [68,69] (Table 1). MET expression is enriched in hepatocytes and in a number of different tumors. The pro-receptor protein MET is cleaved into α and β subunits that form the functional MET receptor by disulfide bonding. MET signaling involves dimerization upon HGF binding and autophosphorylation at Tyr residues, activating PI3K/AKT, RAS/MAPK and STAT pathways. Multiple interactions occur at the intracellular C-terminus of MET. Co-expression of MET and the cell surface mucin MUC20 suggested an association at the C-termini of both glycoproteins that could regulate MET activation [70]. Gangliosides [71] as well as bacterial proteins [72] also have the potential to stimulate or control MET activation. Mutations in the C-terminal Tyr-kinase domain of MET have been found in childhood hepatocellular carcinomas [73].

The extracellular domain of MET contains 11 *N*-glycosylation sites that may be glycosylated and binds the glycoprotein ligand HGF as well as heparin [71,74]. Blocking *N*-glycosylation with tunicamycin resulted in reduced expression levels of MET in glioma tumors in athymic mice [75]. Accumulation of both non-phosphorylated and phosphorylated pro-MET was detected in the cytoplasm upon tunicamycin treatment, suggesting that *N*-glycosylation is critical for transportation of mature MET to the cell surface. In addition, the blocking of *N*-glycosylation could activate phosphorylation independent of HGF [68].

*N*-glycan core fucosylation catalyzed by FUT8 is increased in hepatocellular carcinoma and other cancer types. Using a mouse model for liver regeneration, it was shown that FUT8 contributed to MET signaling and liver regeneration [64,65]. Knockout of the *FUT8* gene in mice led to a decreased response to HGF and delayed hepatocyte proliferation due to decreased phosphorylation of HGF receptors and signaling.

The bisecting GlcNAc of *N*-glycans in hepatocarcinoma HepG2 cells was also found to affect HGF-induced cell signaling [69]. Transfection of cells with the *Gn-T III* gene showed no effect on the expression level of MET or the level of phosphorylated MET. However, increased cell scattering as well as enhanced ERK phosphorylation was detected upon activation with HGF in *Gn-T III* transfectant cells.

Increased α2-6-sialylation of *N*-glycans by ST6Gal I is a marker for colorectal cancer. The MET receptor can be sialylated by ST6Gal I that facilitates the activation of signaling pathways and promotes proliferation and progression of colorectal cancer [76]. Although the mechanism of sialic acid function is still unknown, the terminal sialic acid residue may be required for receptor–protein interactions that guide receptor functions. In mice, ST6Gal I promoted the growth of larger tumors. In human colon cancer cells HCT116, a knockdown of ST6Gal I resulted in the attenuation of MET-mediated JAK2/STAT3 signaling. The loss of α2-6 sialic acid abolished cell motility due to the dephosphorylation of STAT, which led to suppression of MET-mediated cell growth [77]. The α2,3-sialyltransferase ST3GAL4 is involved in the synthesis of sialyl-Lewis^x^ epitopes which act as ligands for selectins and are involved in cancer metastasis. In gastric cancer cells, the expression of ST3GAL4 promoted an invasive phenotype and specific increases in MET activation [78].

MET has potential *O*-glycosylation sites [79] and lectin blots using GalNAc-binding *Vicia villosa* (VVA) lectin showed that GalNAc can be transferred to MET. Furthermore, GALNT2 expression is downregulated in gastric cancer. Suppression of GALNT2 expression in cells with endogenously higher expression resulted in increased cell proliferation and migration. These cells exhibited an increased invasive character in vitro and in vivo. This effect was accompanied by a decreased expression of GalNAc on MET and enhanced MET phosphorylation [79]. This suggests that *O-*glycans have the ability to control receptor activation. However, the question remains of how the other polypeptide GalNAc-transferases expressed in cancer cells are involved in receptor activation and whether GalNAc residues affect receptor conformation and function.

It is likely that GalNAc *O*-glycans in cancer cells are further modified. The most common modification is the synthesis of core 1, Galβ1-3GalNAc, by core 1 β1,3-Gal-transferase C1GALT. This enzyme is highly active in hepatocellular carcinoma and correlates with metastasis and poor survival. Wu et al. [80] overexpressed C1GALT in hepatocellular carcinoma cells which increased cell proliferation. The *O*-glycosylation of MET with GalNAc and core 1 (Tn and T antigens, respectively) was demonstrated using lectins. The opposite effect was seen after suppression of C1GALT expression both in cell cultures and in mice. Thus, downregulation of C1GALT suppressed HGF-induced MET phosphorylation. Since downregulation of C1GALT would expose more GalNAc residues, it is possible that the Tn antigens alone reduce receptor activation while an additional Gal residue has the opposite effect.

### 2.3. Fibroblast Growth Factor Receptor, FGFR

Fibroblast growth factor receptors (FGFR1-4) can be activated through binding FGF isoforms and belong to the family of receptor Tyr kinases that are involved in developmental and pathological processes, angiogenesis and tissue repair (Table 1). FGFR also bind with lower affinity to glycosaminoglycans such as heparan sulfate and heparin oligosaccharides as co-receptors (Figure 2A) [81,82]. FGFR4 is overexpressed particularly in breast cancer [81]. The receptors are decorated with five to eight *N*-glycans [83,84]. The extracellular domain of FGFRI-IIIc produced in CHO cells carried bi- and tri-antennary *N*-glycans having core Fuc and one to three sialic acid residues [82]. Tunicamycin treatment or removal of *N*-glycans by *N*-glycanase increased the binding of FGF2 and heparan oligosaccharides likely because *N-*glycans are sterically hindering and controlling ligand binding.

FGF receptors expressed in baby hamster kidney cells are *N*-glycosylated with oligomannose *N*-glycans that could be removed with *N-*glycosidase F. The de-*N*-glycosylated receptor did not bind ^125^I-labeled FGF, suggesting that the *N*-glycans are essential for FGFR function [83].

FGF19 expression in the terminal ileum is stimulated by excess bile acids in the intestine. The stimulation of FGFR4 by FGF then led to control of bile acid synthesis via cytochrome P450 7A1. The regulator protein β-Klotho binds to FGFR4 carrying mannose-containing *N*-glycans in the ER that are not fully processed to complex-type chains and directs unprocessed FGFR4 to the proteasome of HepG2 cells. As a consequence, only FGFR4 having fully processed *N*-glycans reach the cell surface. Thus, complex *N*-glycans are responsible for FGFR activity and bile acid synthesis from cholesterol [85].

The genetic disease hypochondroplasia can be caused by mutations in the *FGFR3* gene. A mutation of the *N*-glycosylation site Asn328 of FGFR3 was found in patients. This suggests that *N*-glycosylation is crucial for proper FGFR3 function although the mechanism needs to be determined [86]. Another condition, Crouzon syndrome, is associated with abnormal glycosylation of FGFR2. Mutation of Cys278 of FGFR2 increased degradation of the receptor and limited its subcellular localization in osteoblasts. Thus, FGFR2-bound *N*-glycans were shown to be important for intracellular trafficking of the receptor. The lack of *N*-glycans in Cys278 and Asn263 mutants and after tunicamycin treatment of COS-7 cells appeared to promote the ability of receptors to form dimers, indicating that *N*-glycosylation controls receptor function by preventing receptor dimerization [87].

Poly-LacNAc structures were found to regulate FGFR-mediated cell motility of human sperm cells [88]. After removal of poly-LacNAc chains via endo-β-galactosidase, cAMP production and calcium influx in sperm cells was observed. In HEK293 cells, the binding of FGF to FGFR2 was increased after endo-β-galactosidase treatment. This suggested a role of the heavily fucosylated poly-LacNAc chains in controlling FGFR signaling of human sperm cells.

Hung et al. [89] showed that FGFR2 expressed in colon cancer cells is also *O*-glycosylated. Overexpression of C1GALT that synthesizes the cancer-associated T antigen, was shown to modify *O*-glycans of FGFR2 accompanied by enhanced ligand-induced receptor phosphorylation and activation. FGFR2 appeared to carry small amounts of T antigens and sialyl-T antigens. The overexpression of C1GALT led to increased cell migration, invasion and survival as well as tumor growth in immunodeficient mice. The knockdown of C1GALT in colon cancer cells had the opposite effects and showed reduced tumor growth in mice. This phenomenon was similar for HGFR but it is not clear if T (or sialyl-T) antigens on FGFR support receptor function or if Tn/sialyl-Tn antigens reduce receptor activation.

### 2.4. Vascular Endothelial Growth Factor Receptor, VEGFR

Vascular endothelial growth factor receptors (VEGFR1, 2, 3) are Tyr kinases (Figure 2A) and their VEGF ligands promote cell proliferation, migration and differentiation of the endothelium, as well as angiogenesis and development of lymph and blood vessels (Table 1). This is particularly important for the growth of tumors, and thus VEGFR is a target for inhibition for cancer therapy. Both the highly glycosylated, extracellular VEGF-binding domain and the intracellular Tyr kinase signaling domain, possessing many phosphorylation sites, are relatively large [9,90,91].

VEGFR2 is one of the essential receptors involved in the angiogenic signaling pathway. The interaction between VEGF and VEGFR2 triggers dimerization and phosphorylation of the receptor and recruitment of adaptor proteins involved in ERK1/2, FAK and MAPK-mediated cell proliferation, migration and reorganization [9].

Dysregulation of VEGFR2 signaling leads to the formation of abnormal tumor-associated blood vessels, tumor metastasis and resistance to chemotherapies. The extracellular domain of VEGFR2 contains IgG-like repeats with 18 *N*-glycosylation sites, many of which are occupied by complex *N*-glycans [90,92]. In particular, the *N*-glycan at Asn247 regulates receptor activation. Treatment of NIH3T3 fibroblasts with tunicamycin resulted in rapid degradation of VEGFR2. Moreover, *N*-glycans on VEGFR2 were shown to be required for ligand binding and signal transduction [93].

*N*-glycans linked to any of the Asn residues are heterogeneous in structure. The *N*-glycans at Asn247 on cell surface-bound VEGFR2 are generally highly sialylated and fucosylated. The *N*-glycan at Asn145 is also of the complex type but with fewer sialic acid residues, while the *N*-glycan at Asn160 exists as mainly oligomannose chains [91]. Glycans may assume different conformation or different accessibility to enzymes due to their interactions with the peptide environment. This could therefore lead to site-directed glycan processing during biosynthesis. The *N*-glycans at Asn247 carry α2-6 linked sialic acid residues that appear to suppress the VEGFR2-mediated signaling, whereas asialo-glycans facilitate VEGFR2 activation. In ST6Gal I knockout mice, reduced VEGFR2 activation and tumor angiogenesis was observed as well as increased extrinsic and intrinsic apoptosis in the endothelium. It appears that α2-6 sialylation is required for survival of the endothelium through stabilizing glycoproteins, including VEGFR2, at the cell surface [94].

FUT8-deficient mice exhibited an emphysema-like phenotype associated with the decreased expression of VEGFR2. Increased apoptosis and accumulation of ceramide was observed, especially in the lungs. The knockdown of FUT8 in human lung cancer cells also resulted in decreased VEGFR2 expression. Thus, the fucosylation of VEGFR by FUT8 in humans and mice supports receptor function, either by affecting ligand binding, receptor dimerization or appearance at the cell surface [95].

VEGFR2 may also be *O*-glycosylated. COSMC is a required chaperone that stabilizes C1GALT for the synthesis of the T antigen that is overexpressed in many cancers [96]. Interestingly, COSMC is highly expressed in proliferating infantile hemangiomas. Using human umbilical vein endothelial cells (HUVECs), the overexpression of COSMC enhanced VEGF-induced phosphorylation of VEGFR2 and AKT/ERK signaling. COSMC promoted cell proliferation, while knockdown of COSMC via siRNA suppressed endothelial cell growth. The stability of VEGFR2 was found to correlate with the expression of COSMC, and the degradation of VEGFR2 was promoted by knockdown of COSMC. VEGFR2 from HUVEC and hemangioma tissues bound to VVA and *Peanut agglutinin* (PNA) lectins after neuraminidase treatment, suggesting that the receptor is *O*-glycosylated with (sialyl-)T antigens. The overexpression of COSMC in HUVEC supported VEGFR activation and increased VEGFR2 binding to PNA lectin, suggesting that more T antigen was synthesized [96]. However, it is possible that COSMC has functions in addition to T antigen synthesis on VEGFR that could affect receptor activation.

### 2.5. Insulin Receptor and Insulin-Like Growth Factor Receptor

The insulin-mediated and insulin growth factor (IGF1 and IGF2)-mediated signaling in mammals involves a dynamic network of protein ligands and the homologous insulin receptor (INSR) and insulin-like growth factor receptors (IGFR) (Figure 2B). These receptors belong to the Tyr kinase superfamily and can form homodimers or heterodimers. They play critical roles in tumor development and survival and in gene transcription that regulates the uptake and biosynthesis of glucose. The activation of IGFR-mediated signaling pathway was shown to accompany malignant transformation with elevated cell proliferation, survival, and potential for metastasis and tumor angiogenesis [97].

Lectin binding studies showed that the IGFR receptors have 16 to 18 *N*-glycosylation sites (Table 1) that differ in their glycosylation patterns [98]. The *N*-glycans vary from oli-gomannose to hybrid-type to complex chains that have a high content of terminal sialic acids. Interestingly, the glycosylation patterns of INSR and IGFRs have been shown to change during pregnancy [99]. Like other receptors, α1-6-linked Fuc residues are present on the *N*-glycan cores of IGFR1. Gestational changes included a decreased number of Fuc residues in biantennary *N*-glycans and *N*-glycans with α2-6-linked terminal sialic acids on INSR from placentas. A knockdown of FUT8 in trophoblastic cells suppressed cell proliferation, epithelial-mesenchymal transition, migration and invasion by downregulating IGFR1-mediated MAPK and PI3K/AKT signaling pathways [100]. Thus, FUT8 plays an important role in trophoblast proliferation and normal placenta function that depend on IGFR1.

N-linked glycosylation of IGFR1 was shown to be critical for receptor cell surface distribution and crosstalk with the androgen receptor. Synthetic androgen stimulated IGFR1 expression at the plasma membrane. However, inhibition of *N*-glycosylation via tunicamycin in prostate cancer cells led to accumulation of the IGFR1 pro-receptor and reduced the plasma membrane localization of mature IGFR1 [101]. Treatments with tunicamycin, deoxymannojirimycin or castanospermine to inhibit *N*-glycosylation affected receptor cell surface localization in trophoblasts, and IGF1- and IGF2-induced proliferation was attenuated [102].

*N*-glycosylation appears to be responsible for the localization of IGFR1 to the cell membrane [103]. Statins block the biosynthesis of mevalonate in the pathway to cholesterol as well as dolichol-phosphate which is a required intermediate for *N*-glycosylation. Thus, statins and *N*-glycosylation inhibitors cause modifications in IGFR1 glycosylation [102]. This disrupts the ability of IGF1-induced protection against osteogenic differentiation and mineralization of vascular smooth muscle cells (vascular calcification) through decreased IGFR processing and cell surface transportation. Modification of IGFR1 glycosylation also downregulated AKT and MAPK signaling pathways [103,104]. The need for *N*-glycosylation in IGFR1 and INSR receptor proteolytic processing and membrane localization was also demonstrated in HEK293 cells lacking the catalytic subunits of *N*-oligosaccharyltransferase [105].

Receptor *N*-glycans may carry linear or branched *N*-acetyllactosamine chains with a number of sialic acids, Fuc and Lewis epitopes. Loss of the β1,6-GlcNAc-transferase GCNT2 that synthesizes the I antigen branches of N-acetyllactosamine chains was observed in melanomas, and a knockdown of GCNT2 activated the IGFR1-mediated signaling pathways and Tyr phosphorylation upon IGF1 induction in melanoma cells. In contrast, overexpression of GCNT2 resulted in decreased phosphorylation of FAK and ERK1/2 and attenuation of IGF1-induced cell proliferation [106]. Thus, the additional branches in complex *N*-glycans introduced by GCNT2 appear to interfere with receptor function.

*O*-glycans on the INSR and IGFR1 receptors have not been detected. However, overexpression of polypeptide GalNAc-transferases may force protein *O*-glycosylation that does not occur at normal expression levels. GALNT2 overexpression in neuroblastoma cells resulted in increased *O*-linked GalNAc residues on IGFR1 recognized by VVA lectin. The increased GALNT2 expression levels reduced receptor dimerization. In contrast, GALNT2 knockdown enhanced IGF1-induced cell proliferation, migration and invasion [107]. Tumor growth in nude mice was much higher in GALNT2-deficient melanoma cells, thus, receptor-bound GalNAc-*O*-glycans may block receptor function and the malignant phenotype.

Neuraminidases are lysosomal hydrolases that can also be part of cell surface protein complexes where they associate with a number of cell surface receptors such as INSR. Desialylation of IGFR1 of arterial smooth muscle cells by neuraminidase Neu1 increased proliferation indicating that terminal α2-3/6-linked sialic acid residues control receptor activation in response to insulin and IGF2 [108,109,110]. Desialylation also sensitized INSR in rat skeletal myoblast cells and induced cell proliferation and tissue regeneration in response to low concentrations of insulin. Therefore, it appears that receptors are more sensitive to insulin when sialic acid is lacking.

### 2.6. Receptors for Neurotrophic Factors

Neurotrophic factors include neurturin and neurotrophins NGF3, NGF4, GDNF that activate their respective receptors expressed on neuronal cell membranes and stimulate survival, neurite outgrowth, synapse formation and other neuronal functions (Table 3, Figure 2C) [111]. RET (REarranged during Transfection), TrkA, TrkB, TrkC are Tyr kinase receptors. NGFR p75 does not have Tyr kinase activity but has homology to TNFR and can induce both neuronal survival and cell death [112]. Polysialic acids (PolySia) are often found attached to *N*-glycans in neuronal cells. These highly charged glycans are thought to be anti-adhesive and critical for development and maintenance of the nervous system. PolySia specifically bind neurotrophins, growth factors and neurotransmitters, thereby regulating their processing and functions [113].

The p75 neurotrophin receptor NGFR is an apical cell surface glycoprotein that binds relatively non-specifically NGF, BDNF, NTF3 and NTF4 and regulates the circadian rhythm and tissue regeneration [112]. NGFR can form dimers with other receptors (e.g., TrkA) and acquire high affinity for their ligands. P75 is a TNFR superfamily 16 protein that can induce apoptosis. It has only one *N*-glycosylation site but it carries core 1 *O*-glycans at the stalk near the transmembrane domain [114]. The *O*-glycans, but not the *N*-glycan, appear to be critical for apical sorting of the p75 receptor and may play a role in receptor conformation.

RET is a transmembrane receptor Tyr kinase that is activated by a complex consisting of a soluble GDNF family ligand and a GPI-anchored coreceptor GFRα of the GDNF receptor family [115] where the GPI anchor is linked to the C-terminus of the receptor and thus anchors the protein in the cell membrane. The GFRα 1, 2, 3, 4 coreceptors are extracellular glycoproteins with 1 to 4 *N*-glycosylation sites that participate in binding RET, NGF and GDNF. RET can be cleaved by caspase into an extracellular N-terminal α-domain with four cadherin domains that share adhesion functions with cadherin and the transmembrane domain. The intracellular β-domain has two caspase cleavage sites [116] and contains the Tyr kinase domain that can induce apoptosis. RET functions in proliferation through MAPK and AKT pathways and contributes to growth of neuroendocrine cancer. Multiple mutations in both domains are associated with pathology.

In order to study the role of bisected *N*-glycan structures, Gn-T III was overexpressed in rat pheochromocytoma PC12 cells. Neurite outgrowth and cell growth were depressed in transfected cells stimulated with NGF, and no Trk phosphorylation and dimers were detected [117]. Binding of *Phytohemagglutinin*-E (PHA-E) lectin that recognizes bisected *N*-glycans on Trk was increased in Gn-T III transfected cells. This clearly indicated that receptor function is suppressed by bisected *N*-glycans and the presence of highly branched *N*-glycans support receptor function. TrkA from PC12 cells is a substrate for Gn-T V that synthesizes the GlcNAcβ1-6Man linkage recognized by *Phytohemagglutinin-*L (PHA-L) lectin. The overexpression of the Gn-T V led to increased PHA-L binding, receptor activation and phosphorylation [118].

Trk A, B, C receptors are relatively specific for their ligands and have IgG-like, heavily *N*-glycosylated extracellular domains [119]. NTRK1 (TrkA) has 13 *N*-glycosylation sites, 6 of them (including Asn188, Asn 281) carrying oligomannose *N*-glycans were clearly identified in the crystal structure of the extracellular domain [120]. TrkA can form a functional heterodimer with other receptors. Upon dimeric NGF ligand binding, TrkA undergoes homodimerization, autophosphorylation and activation. It promotes proliferation of neuroblastomas. *N*-glycosylation was found to be important for receptor localization to the cell surface and for promoting neuronal differentiation of PC12 cells [121]. TrkA can also undergo nonenzymatic glycation by the addition of glucose to the ε-amino group of Lys, decreasing its ligand affinity for NGF [122].

Woronovicz et al. [123] showed that TrkA is regulated by Neu1, but not Neu2 or Neu3. TrkA has sialylα2-3 termini that are recognized by *Maackia amurensis* (MAA) lectin and can be cleaved by Neu1. The expression of Neu1 was stimulated by NGF in PC12 cells that express TrkA, and neuraminidase inhibitors including Tamiflu blocked TrkA expression, as well as neurite outgrowth of PC12 cells after NGF stimulation. Mouse primary cortical neurons that express NTRK2 (TrkB) also produced neuraminidase when stimulated with BDNF, and Tamiflu blocked TrkB phosphorylation [123]. This suggests that sialic acid residues of Trk control receptor activation. TrkA also interacts with GM1 and this activates the MAPK pathway and receptor phosphorylation, and enhances neurite outgrowth from mouse neuroblastoma cells. The interactions between GM1 and the extracellular domain of TrkA require the oligosaccharide moiety of GM1, and sialic acid is an essential component which can be cleaved by Neu3 [124]. Thus, sialic acid appears to have the dual function of blocking receptor activation when linked to protein-bound glycans but is required for activation through binding of GM1.

Ten of the 11 *N*-glycosylation sites of TrkB were shown to be occupied by *N*-glycans [125]. TrkB binds BDNF, NGF3 and NGF4 and plays a role in learning and memory. Mutations in TrkB are found in congenital aganglionosis (Hirschsprung disease) [126]. A number of patients with congenital heart disease were shown to have mutations in the *TrkC* (*NTRK3*) gene [127]. TrkC Tyr kinase binds preferably NGF3 and has 13 *N*-glycosylation sites with yet unknown roles. It does not appear that any of the Trk receptors are *O*-glycosylated.

## 3. Transforming Growth Factor-Beta Receptor, TGFBR

Transforming growth factor-β receptors (TGFR, TβR) are Ser/Thr kinase receptors that regulate proliferation and differentiation, tissue repair and immune responses. They signal by activating SMAD pathways, mainly through binding TGFβ family members TGF1, 2 or 3 [128] (Table 4). TGFβ-induced signaling plays a major role in the development of cancer and TGFR mediates tumor growth by inducing EMT [129]. TGFR can form homodimers or heterodimers (TGFR I/TGFR II) in a receptor complex of four receptors and the cytokine dimer with different affinities for their ligands (Figure 3). The cytoplasmic Ser/Gly-rich GS domain precedes the Ser/Thr kinase domain and becomes phosphorylated upon ligand binding which then propagates signaling. The Ser/Thr kinase domain is relatively large, and mutations in this domain are associated with disease. Three types of TGFR are known with TGFR I and TGFR II having one and three *N*-glycosylation sites, respectively, and a truncated TGFR III having five *N*-glycosylation sites, two of which carry glycosaminoglycans with Xylose linked to Ser in the extracellular domain. In contrast to TGFR I and II, the large extracellular, secreted domain of TGFR III can function as coreceptor and bind to TGFβ [130].

Glycosylation can function in a number of ways, from controlling protein conformation to interactions with carbohydrate-binding molecules and glycolipids (Figure 3) [61]. The *N*-glycosylation of TGFR II is essential for the receptor transportation to the cell surface [131]. Inhibition of *N*-glycosylation with tunicamycin and mannosidase I inhibitor kifunensine in A549 lung cancer cells prevented TGFR II receptor cell surface distribution and reduced SMAD2 phosphorylation and signaling. Although TGFR II has three *N*-glycosylation sites, only conserved residues Asn70 and Asn94 appear to carry *N*-glycans. TGFR II from gastric carcinoma cells or HEK293 cells had both complex and oligomannose structures on the plasma membrane-bound receptor. The TGFβ sensitivity appeared to be reduced after inhibition of *N*-glycosylation [132].

*N*-glycan core fucosylation was found on several types of growth factor receptors including TGFR [61,133], and a number of studies showed the importance of FUT8. FUT8 expression is upregulated upon TGFβ treatment of human breast cancer cells which correlates with EMT of human mammary epithelial cells MCF-10A in vitro and with the invasive ability of breast cancer cells in nude mice [134]. Knockdown of core fucosylation with shFUT8 suppressed the invasiveness of metastatic breast cancer cells and EMT in MCF-10A cells. In renal tubular cells, the phosphorylation of SMAD2/3 and TGF-β-SMAD2/3 signaling [135] was reduced by FUT8 siRNA. Similar effects of fucosylated TGF-β receptors were observed in colon cancer cells [136]. Thus, blocking the core fucosylation could be a strategy to attenuate EMT by suppressing the TGFβ/SMAD signaling pathway [133,137].

In mice, core fucosylation of *N*-glycans of TGFR I and II is required for receptor cell surface expression and for interactions with TGFβ [138]. In FUT8 knockout mice, TGFβ-induced signaling and SMAD phosphorylation were found to be defective, and the gene expression of matrix metalloproteinases (MMP) encoding MMP12 and 13 was upregulated [134,137,139]. *FUT8* gene knockout mice experienced progressive emphysema. The reduction of TGFβ-mediated signaling could be restored by re-introduction of the *FUT8* gene [140]. Clearly, the Fuc residue linked to the core of *N*-glycans plays a major role in TGFR signaling and in cancer progression, and could be a target for anticancer therapy. However, the molecular mechanisms by which this Fuc residue exerts its effects have not been determined. It is possible that it directly binds to amino acids of the receptor protein backbone. However, it may also influence the conformation of the *N*-glycan, its exposure into the extracellular space, or the interaction between sugar residues and the peptide and thus the interactions between two TGFR molecules.

TGFR II in cancer cells has also been shown to be modified by α1-3-linked Fuc that influences TGFβ-induced phosphorylation [136,141]. The expression of sialyl-Lewis^x^ and sialyl-Lewis^a^ epitopes, as well as α1,3-Fuc-transferases FUT 3 and FUT6, also appeared to be critical for activating TGFR I [136].

Tetraantennary *N*-glycans include GlcNAcβ1,6-linked branches synthesized by Gn-T V. Gn-T V appears to control the number of Gal residues exposed at cell surfaces that are therefore available for binding galectins. The enzyme is highly expressed in cancer [14] and acts on TGFR I and II. In mice, Gn-T V regulates the TFGβ response in hepatic stellate cells and appears to be required for EMT [142]. In contrast, overexpression of Gn-T III showed the opposite effect and inhibited TFG-β-induced cell motility and EMT process in both human and mouse breast cancer cells [143]. GCNT2 is another branching β1,6-GlcNAc-transferase that has a significant potential to alter the overall structure of the glycan. The expression of GCNT2 is high in breast cancer and regulates cancer cell invasion and migration. Downregulation of GCNT2 expression prevented the TGF-β-induced EMT process, as well as migration and invasion of breast cancer cells [144]. This enhancing effect of GCNT2 on TGFR function is in contrast to that on IGFR1-mediated signaling [106]. During EMT, ST6Gal I expression is upregulated. However, a knockdown of ST6Gal (but not α3-sialyltransferase ST3Gal4) suppressed TGFβ-induced EMT in GE11 epithelial cells [145], indicating that the sialylα2-6 linkage controls TGFR activation.

TGFR may also be *O*-glycosylated. Wu et al. [146] showed that TGFR II from breast cancer cells was *O*-glycosylated at Ser31 and possibly also at Thr39. GALNT4 was shown to be the enzyme that transfers GalNAc to both TGFR I and II and blocks heterodimer formation. Downregulation of GALNT4 resulted in enhanced cancer cell migration and invasion via EMT, while GALNT4 overexpression showed the opposite effect. GALNT4 knockout resulted in increased dimerization and signaling of TGFR. Receptor mutants lacking Ser31 showed increased TGF-induced signaling in breast cancer cells. This suggests that GalNAc residues of TGFR control receptor function. Depending on the cell type and the GTs expressed, GalNAc could be further modified to more complex *O*-glycans that would potentially regulate dimerization and signaling [146].

## 4. Death Receptors

Death receptors belong to the tumor necrosis factor (TNF) receptor superfamily and are essential regulators of cell death and survival, immune responses and tumor growth (Table 5). These glycosylated type I plasma membrane proteins function in the initiation of apoptosis (Figure 4) pathways which were found to be glycosylation-dependent [61]. In mammals, apoptosis occurs via intrinsic and extrinsic pathways [147]. Intrinsic apoptosis pathways are mitochondria-mediated and involve cytochrome c, Apaf-1 and caspases, and are regulated by pro-apoptotic Bcl-2 family members [148]. These intracellular proteins may be modified by *O*-GlcNAc attached to Ser/Thr residues that play a role in regulating phosphorylation and function [15]. The extrinsic apoptosis pathways are mediated by the interactions of apoptosis-inducing ligands with the ectodomain of cell surface glycoprotein death receptors and are further propagated by the cytoplasmic death domain. The ligands are TNF family members that are also glycosylated, including TNFα, TRAIL and Fas ligand (FasL). Upon ligand–receptor interaction, receptors undergo trimerization, leading to recruitment of intracellular adaptor proteins and the assembly of death-inducing signaling complex (DISC) in the cytoplasm, activation of protease caspase 8 and signaling pathways [147]. Receptor-bound glycans can regulate receptor protein folding and plasma membrane expression and binding to their ligands (Figure 4). Furthermore, glycans mediate control over oligomerization leading to signaling, and to interaction with carbohydrate-binding proteins at the cell surface that can influence receptor functions [149,150]. There are also decoy receptors that lack the cytoplasmic death domain and cannot induce apoptosis but compete with death receptors for ligand binding. All of these receptors have varying affinities for TNFα and TNFα-like ligands and regulate cell death and survival.

### 4.1. Fas Glycosylation

Fas (CD95) is a member of the TNFR superfamily and is a glycosylated membrane receptor that contains two *N*-glycosylation sites in the extracellular domain [151,152]. Fas-mediated apoptosis pathways are triggered by binding to Fas ligand (FasL) and also by an anti-CD95 antibody that may link subunits together (Figure 4). The two *N*-glycans of Fas are close to the ligand binding site. *N*-glycans have been shown to regulate the expression level and secretion of FasL [13,153]. The sensitivity of Fas to ligand-induced apoptosis was found to be glycosylation-dependent via galectins that bind to terminal Gal residues [149]. Galectins also interact with Fas intracellularly.

Tunicamycin treatment of HeLa cells showed a reduction of caspase 8 activation at DISC. When expressed in SHIP-1-expressing Jurkat cells, Fas appeared to be *O*-glycosylated [154] but *O*-glycosylation of Fas may depend on the cell type and the role of *O*-glycans remains to be examined. A possible *O*-glycosylation site (at Thr29) remains to be confirmed.

Sialylation is increased in many proliferating cancer cells [155] and appears to reduce the sensitivity of death-receptor-mediated apoptosis. A possible mechanism is the repulsion of negative charges of receptor glycans that prevent oligomerization. B-lymphocytic cells expressing less sialylα2-6 residues were found to be more sensitive to Fas-mediated apoptosis. Desialylation of Fas glycans via α2-3/6/8-neuraminidase from *Vibrio cholerae* enhanced its susceptibility to Fas-mediated apoptosis [156,157]. Fas *N*-glycans are substrates for ST6Gal I, and α2-6 sialylation of Fas led to an inhibition of Fas-mediated apoptosis. This seems to be selective for Fas function and not for function of TRAIL receptors [158]. In colon carcinoma cells, knockdown and overexpression of ST6Gal I indicated that α2-6 sialylation blocked receptor oligomerization and signaling induced by both FasL and anti-CD95 antibody. Thus, highly α2-6 sialylated Fas does not associate with FADD in the cytoplasmic domain and thus blocks DISC formation and signaling, limiting apoptosis. The mechanism by which this occurs is likely the control of receptor conformation and functional trimer formation. It is important to understand the regulation of cell death to develop anti-tumor therapies and to find treatments for hyperactive immune responses.

### 4.2. TNF/TRAIL Receptors

Five more apoptosis-inducing receptors, DR3, DR4, DR5, DR6 and TNFR1A, of the TNFR superfamily activate and regulate apoptosis in various cell types. TNFα can activate TNFR1 (TNFRSF1A) and lead to either cell survival (by binding TRADD) or cell death (by binding FADD) that involves internalization of the receptor complex II (Figure 4). All of these receptors are glycoproteins with the Cys-rich extracellular domain carrying up to six *N*-glycans (Table 4). DR4 (TNFRSF10A) and DR5 (TNFRSF10B) bind TRAIL and TNFα [147] and can trigger cell death primarily in cancer cells. DR6 functions in neuronal cells and is S-palmitoylated and extensively glycosylated. The *N*-glycans are responsible for directing DR6 to the cell membrane. DR6 also carries several *O*-glycans in the stalk region above the cell membrane from Thr212 to Thr254, and the extension of *O*-glycans could be reduced by GalNAcα-benzyl treatment [159]. The membrane-bound forms can be proteo-lytically cleaved and found as soluble ligands in the extracellular space as decoy receptors [160].

Glycosylation plays critical roles in controlling the sensitivity of TRAIL-induced apoptosis [150]. DR4 contains one *N-*glycosylation site while DR5 does not have *N*-glycans [161], but both DR4 and DR5 are *O*-glycosylated in humans and other species [162]. The ligands for TNFR include TNFα which is a type II membrane protein that regulates both proliferation and apoptosis. It is produced mainly by macrophages and can be proteolytically cleaved from the membrane to form the extracellular soluble TNFα ligand.

*N*-glycosylation of DR4 is not necessary for its expression, cell surface distribution and ligand binding. However, it was shown to play a role in TRAIL-induced apoptosis by enhancing receptor clustering and DISC formation, both in humans and mice. The inhibition of *N*-glycosylation by tunicamycin led to enhanced TRAIL-induced apoptosis in human colon cancer cells. It is possible that the mechanism of tunicamycin involves upregulation of DR5 and inhibition of EGFR function [163].

In mice, inhibition of *N*-glycosylation via tunicamycin sensitized cells to TRAIL-induced apoptosis through enhanced ligand binding and receptor oligomerization, resul-ting in formation of DISC and caspase 8 activation. The inhibitory role of *N*-glycans in TRAIL-induced apoptosis was confirmed with *N*-glycosylation site mutants of the murine TRAIL receptor [10].

*O*-glycosylation has also been shown to contribute to TRAIL-induced apoptosis [162,164,165]. DR5 has Ser/Thr-linked *O*-glycans in the Cys-rich domains 2 and 3 that have multiple adjacent Ser and Thr residues. Mass spectrometry analyses showed that the *O*-glycans of DR5 expressed in CHO cells consist of one to four Galβ1-3GalNAc chains (T antigen) with up to two sialic acid residues. These *O*-glycans appear to be critical for the clustering of DR5 and apoptosis induction. Inhibition of *O*-glycan extension by GalNAcα-benzyl in Colo205 colon cancer cells resulted in decreased T antigen and increased GalNAc (Tn antigen). COSMC is required for the formation of core 1 *O*-glycan (T antigen). Mutations of the *cosmc* gene caused the production of truncated *O*-glycans (Tn and STn antigens). Thus, *O*-glycan truncation led to attenuated apoptotic signaling and caused reduced sensitivity to TRAIL. A direct function of *O*-glycans was shown in colon cancer cells deficient in the chaperone COSMC required for *O*-glycan core 1 synthesis. These cells express Tn (GalNAc-) and sialyl-Tn antigens and have a low sensitivity to TRAIL. By transfecting the *cosmc* gene into these cells, TRAIL sensitivity was restored, likely by improving the stability of DR4/5 and ligand-induced receptor oligomerization [164]. This suggests that *O*-glycans that are further processed to core 1, and possibly to core 2 structures, are required for efficient apoptosis to occur.

The enzyme polypeptide GalNAc-transferase (GALNT14) appears to be mainly responsible for *O*-glycosylation, and knockdown of GALNT14 with siRNA reduced TRAIL sensitivity and TRAIL-induced apoptosis, while overexpression increased sensitivity in several cancer cell types. Treatment of cancer cells with GALNT14 siRNA reduced the sensitivity of cells to TRAIL-induced apoptosis. Increased apoptosis after GALNT14 transfection was seen in DR4- and DR5-overexpressed HEK293 cells but not for Fas and TNFR1 that do not carry *O*-glycans. The expression of GALNT14 correlated with Apo2L/TRAIL sensitivity in several cancer types. However, in other cell types, the expression of GALNT family members (i.e., GALNT3) other than GALNT14 also correlated with TRAIL sensitivity [162]. Thus, the relationship between *O*-glycan biosynthesis and apoptosis needs to be considered in the context of specific cell types. GALNT14 may be responsible for ligand-induced translocation of DR4 and DR5 and the formation of DISC, as well as the activation of caspase 8, resulting in cleavage of Bid, caspase 9 and 3 [162,165]. It is possible that DR4 and DR5 are specific or selective acceptor substrates for GALNT14.

In addition to extracellular glycans, DR4 also has either an *O*-GlcNAc residue or a phosphate group linked to Ser424 in the cytoplasmic death domain [15]. After stimulation of human cancer cells with TRAIL, *O*-GlcNAc was found in DR5 but not in DR4 [166]. The Ser424 mutation in DR4 receptor prevented the addition of *O*-GlcNAc and led to TRAIL resistance. The *O*-GlcNAc-Ser424 modification of DR4 is thus essential for the induction of apoptosis (Figure 4).

Specific glycan structures of TNFR1 play a critical role in regulating the dynamic processes of cell survival and death [162,167,168]. Tunicamycin and *N*-glycosidase treatments showed that TNFR1 is *N*-glycosylated in human HEK293 and mouse microglial cells. In addition, mutations in the Asn151 and Asn202 *N*-glycosylation sites of TNFR1 resulted in reduced ligand binding affinity and susceptibility to TNFα-stimulated NF-κB signaling in microglial cells. Therefore, *N*-glycans facilitate the activation of microglial cells during inflammation of the murine central nervous system [169].

Regulation of apoptosis appears to depend on the structures of complex *N*-glycans. Knockdown of β1,4-Gal-transferase 1 expression inhibited TNFα-induced apoptosis in Schwann cells. In contrast, β1,4-Gal-transferase 1 overexpression showed enhanced apoptosis through activation of ERK, JNK and MAPK signaling pathways [170]. Thus, β1,4-Gal-transferase I is closely linked to TNFα-induced apoptosis [168,169].

Most cell surface receptors are fucosylated and this has been suggested to affect the biological functions of receptors [171]. Colon cancer cells HCT116 have a mutation of GDP-mannose-4,6-dehydratase (GMD), an enzyme required for the biosynthesis of GDP-Fuc, the essential donor substrate for Fuc-transferases [172]. DR4 was found to be fucosylated but not DR5 that lacks *N*-glycans. However, both DR5 and DR4 functions were affected by the lack of GMD [172]. This suggests that the Fuc residues of *O*-glycans may be important for receptor activation. While GMD deficiency did not affect the formation of DISC or the recruitment and activation of caspase 8, the formation of secondary FADD-dependent complex II was prevented. Rescue of fucosylation by transfection of the *GMD* gene stimulated apoptosis and reduced tumor growth in athymic mice. Thus, GMD deficiency resulted in the escape of immune cell-mediated tumor surveillance by disrupting TRAIL signaling.

Colon cancer cells COLO 205 highly express α3/4-Fuc-transferase FUT3 and α3-Fuc-transferase FUT6 and are TRAIL-sensitive. However, fucosylation appears to affect DR4 and DR5 differently. Low levels of FUT3 and FUT6 correspond to DR5 TRAIL insensitivity but still allow DR4-mediated apoptosis [173]. The knockdown of FUT6 in several cancer cell types was found to reduce TRAIL-induced caspase 8 activation at DISC, thereby desensitizing cells to TRAIL [162]. Overexpression of FUT3 and FUT6 restored TRAIL sensitivity via DR5. This study suggested that fucosylation plays a critical role in TRAIL signaling. However, it remains to be shown if large *O*-glycans of DR5 indeed carry Fuc residues or if this regulation is through other fucosylated glycoconjugates.

Sialylation was also found to regulate the function of TNFR1 (TNFRSF1A) death receptor [174], and the receptor is a substrate for ST6Gal I in monocytic cells. ST6Gal I expression is upregulated in many cancer types, including in gastric cancer organoid-derived monolayers [175]. Overexpression of ST6Gal I in macrophages prevented TNFα-activated TNFR1-mediated apoptosis, while TNFR1-mediated apoptosis could be restored by treatment with neuraminidase and ST6Gal I shRNA transfection. Upon prolonged overexpression of ST6Gal I in pancreatic and ovarian cancer cells, resistance to TNFα-induced apoptosis developed, preventing caspase 8 and 3 activation [167]. In contrast, removal of α2-6-linked sialic acid residues via α2-3,6,8,9-neuraminidase from *Arthrobacter ureafaciens* enhanced the activation of caspases [175]. Thus, α2-6-sialylation of *N*-glycans acts as a regulator of TNF-induced apoptosis.

## 5. Conclusions

Cell surface-bound receptor glycoproteins regulate cell proliferation and cell death throughout the body. Growth factors, cytokines and other inflammatory stimuli have the potential to modify glycosylation by changing the expression of GTs that act on receptors. Receptor glycosylation has to be considered in context with the specific cells they occupy, with the intra- and extracellular environment, and, importantly, with the membrane environment. In addition, each glycosylation site may have a unique role in regulating receptor functions. The individual glycosylation sites are not equivalent, and only a fraction of these sites actually carry glycans. The co-receptors and ligands carry glycans that may also influence their biological activities. The roles of glycans are complex and include membrane expression of receptors, protein conformation, oligomerization, ligand binding and induction of the signaling cascades. The ligand binding site may contain only a few glycans but the protein around the ligand site has most of the glycans that ensure proper conformation, stability and protease resistance of the receptor. It is very difficult to determine a specific role of individual glycan structures, as glycan biosynthesis is dynamic, structures are heterogeneous, and they vary considerably among receptors in different cell types. Furthermore, glycosylation is dynamically changing, according to the state and context of the cell or tumor and can be influenced by the presence of regulators. Many data have been collected by experiments that knockdown or overexpress glycosyltransferases in different cell types. The presence of *N*-glycans, their branching patterns, fucosylation and sialylation have consistently been shown to be important. A common method to examine the role of *N*-glycosylation is enzyme inhibition. However, cells treated with inhibitors alter their total *N*-glycome, not only that of one specific receptor. In addition, tunicamycin can induce ER stress leading to cell death. Mutation of *N*-glycosylation sites is more specific, but does not consider the structural details of the *N*-glycan. Enzymatic removal or substitution by Fuc or sialic acid residues on cell surfaces also lacks specificity for a single type of receptor. The roles of *O*-glycans appear to differ between the first sugar (GalNAc) and subsequent sugars, which is interesting but poorly understood. Thus, the story of glycan functions is extremely complex but in light of obvious effects of glycans on receptor functions, this field needs much more attention. The specific glycoforms present on receptors and their ligands should be examined to assess the effects of individual sugars and linkages in the glycan chains on receptor conformation and activation. This knowledge is important in order to control tissue repair, immune responses and tumor growth through modifications of receptor-bound glycans.

## Figures and Tables

**Figure 1 cells-10-01252-f001:**
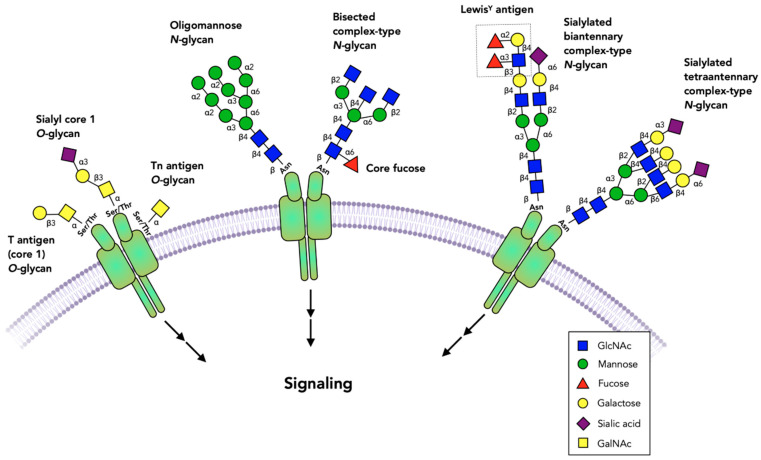
Glycans commonly found on cell surface receptors. The extracellular domains of receptors are glycosylated, and specific glycans have been shown to either block or activate receptor dimerization or activation. Virtually all receptors have multiple *N*-glycans that include oligomannose, complex or bisected structures. Fucose bound to the *N*-glycan core or to *N*-acetyllactosamine chains and sialic acid are among the terminal sugar residues. Some receptors have *O*-glycans that are commonly comprised of simple GalNAc (Tn antigen), core 1 (T antigen) and sialyl-T antigen structures. Not all possible glycosylation sites are occupied and the role of these glycans may vary between occupied sites, cell types and the specific receptors.

**Figure 2 cells-10-01252-f002:**
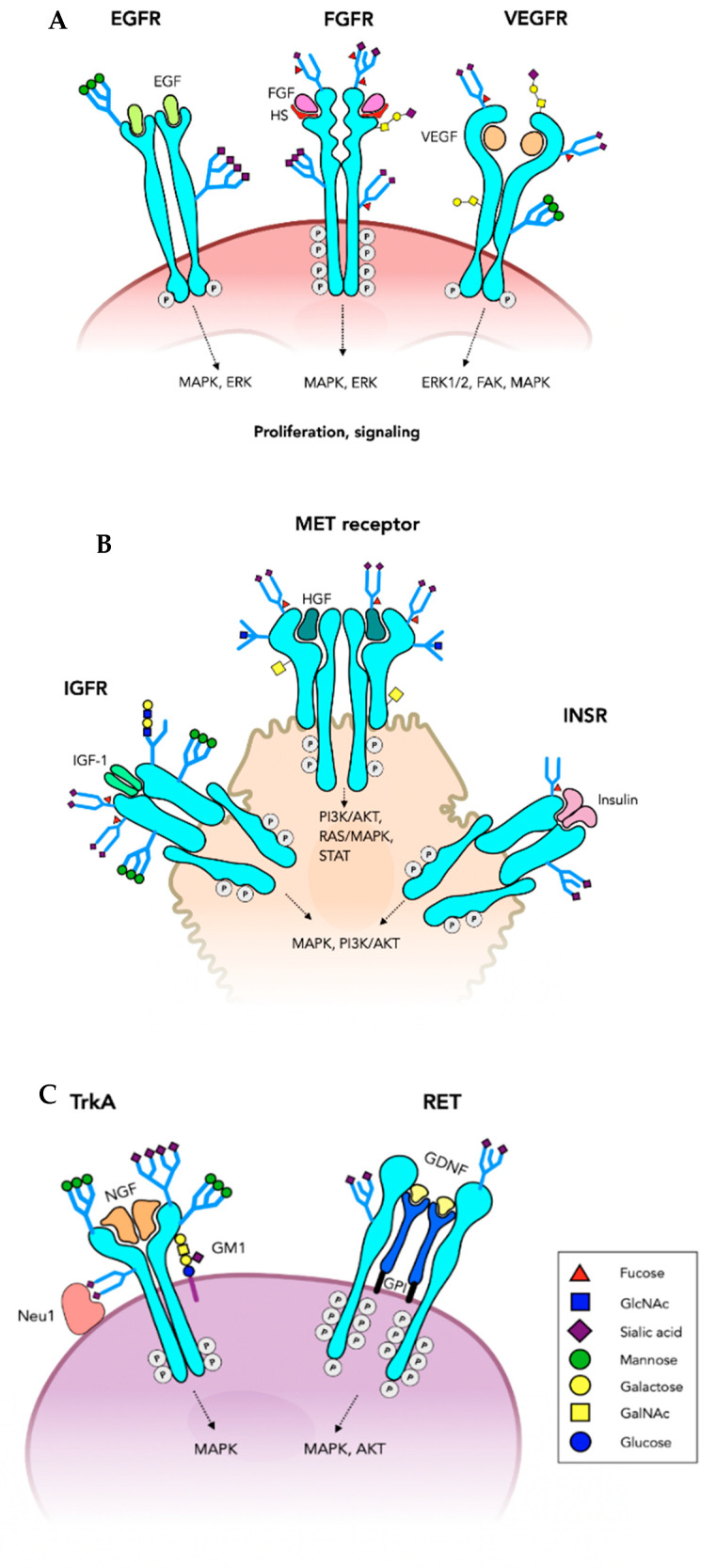
Tyrosine kinase receptors. Growth factor receptors have an extracellular domain that binds a growth factor and a cytoplasmic Tyrosine kinase domain (oncoprotein) that is essential for the signaling through a number of pathways that promote cell proliferation. Many of these receptors are of critical importance in tumor growth. Ligand binding induces receptor dimerization and activation of the Tyrosine kinases, phosphorylation of the cytoplasmic domain and induction of signaling pathways. (**A**) Only one of the *N*-glycans on each of the epidermal growth factor receptor (EGFR) subunits is shown. Fibroblast growth factor receptor (FGFR) also has multiple *N*-glycans and can carry *O*-glycans and have glycosaminoglycan coreceptors (heparin, HS). Vascular endothelial growth factor receptor (VEGFR) has core 1 *O*-glycans and is highly *N*-glycosylated. (**B**) The insulin binding growth factor receptor (INSR) and insulin-like growth factor receptor (IGFR) are *N*-glycosylated glycoproteins. MET receptor binds hepatocyte growth factor (HGF) glycoprotein and is both *N*- and *O*-glycosylated. (**C**) Tyrosine kinase growth factor receptors expressed on neurons include TrkA, TrkB, TrkC and RET involved in cell proliferation, neurite outgrowth and other functions. TrkA has multiple *N*-glycans and interacts with its nerve growth factor (NGF) ligand, as well as ganglioside GM1 and neuraminidase Neu1 on the cell membrane. The RET dimer requires coreceptors that are glycosylphosphatidylinositol (GPI)-anchored at the membrane and bind glial cell-derived neurotrophic factor (GDNF).

**Figure 3 cells-10-01252-f003:**
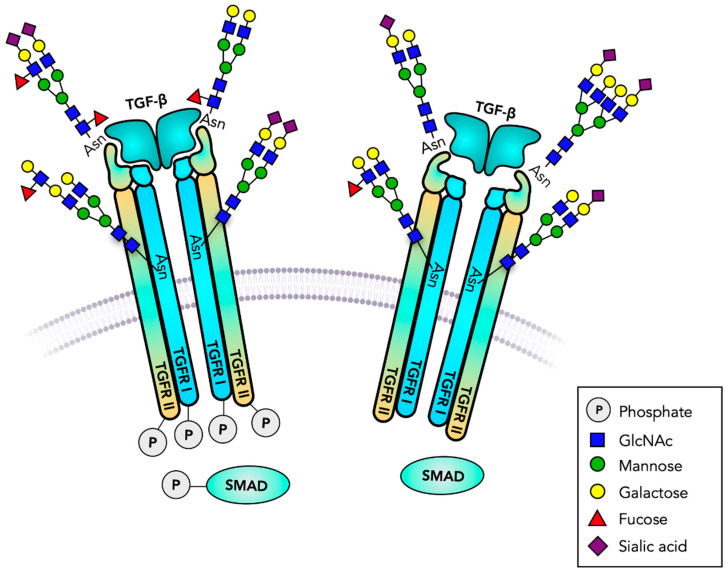
TGF-β receptors are serine/threonine kinases. Transforming growth factor receptors (TGFR) are heterodimeric transmembrane receptors that form tetrameric transmembrane receptors. Binding to TGF-β initiates intracellular serine/threonine phosphorylation, cell proliferation, differentiation and many other biological responses. The signaling cascades and gene regulation involve SMAD phosphorylation. TGFR has several functionally important *N*-glycans, and may carry glycosaminoglycans and *O*-glycans. A Fuc residue at the *N*-glycan core plays a major role in receptor function. Truncated, GPI-anchored, receptor analogs regulate the binding of TGF-β.

**Figure 4 cells-10-01252-f004:**
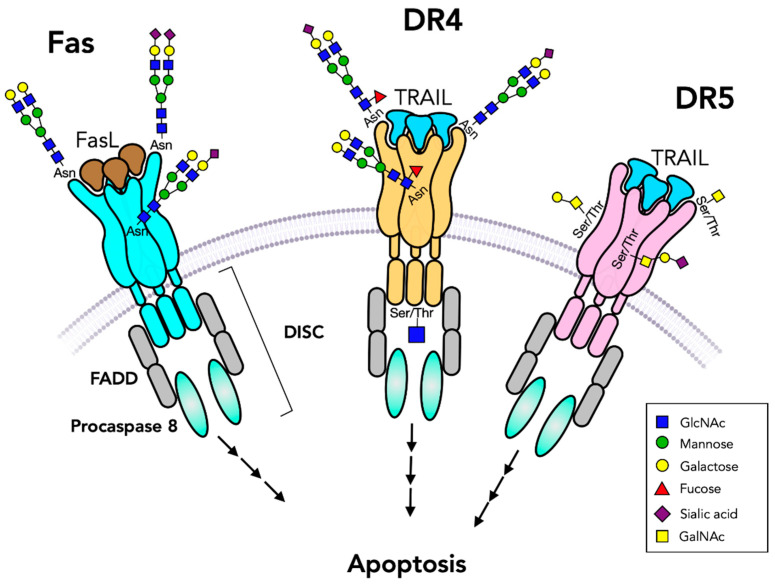
Death receptors initiate apoptosis pathways. Fas and TNFα-related apoptosis-inducing ligand (TRAIL) receptors bind to a trimeric ligand (FasL or TRAIL) and form trimeric receptors that initiate caspase cleavages and apoptosis signaling pathways through Fas-associated protein with death domain (FADD) and death-inducing signaling complex (DISC). Fas has two *N*-glycans while DR4 has only one *N*-glycan and DR5 has no *N*-glycans. Both DR4 and DR5 have *O*-glycans. *O*-GlcNAc residues may be present on the cytoplasmic domain. TRAIL receptors are also controlled by their *N*-glycans. The presence of unmodified GalNAc residues (Tn antigen) leads to attenuation of receptor functions.

**Table 1 cells-10-01252-t001:** Tyrosine kinase receptors.

Protein Uniprot No.	Extracellular Domain	*N*-glycosylation Sites	Cytoplasmic Domain	Function
EGFR P00533	binds EGF, TGFα, MUC1, Neu3, at least 4 signaling cascades	13	Tyr-kinase	cell migration, proliferation
ErbB2 P04626		7	Tyr-kinase	proliferation
ErbB3 P21860		10	Tyr-kinase	proliferation
Erb4 Q15303		11	Tyr-kinase	proliferation
Hepatocyte GFR,MET P08581	Siaα6, O-glycans, binds MUC20, HGF	11	Tyr-kinase	proliferation
FGFR1 P11362	oligoMan, complex N-glycans, binds FGF	8	Tyr-kinase	proliferation
FGFR2 P21802	poly-LacNAc, O-glycans, binds FGF	8	Tyr-kinase	proliferation
FGFR3 P22607	binds FGF	6	Tyr-kinase	proliferation
FGFR4 P22455	binds FGF	5	Tyr-kinase	proliferation
VEGFR1 P17948	binds VEGF	13	Tyr-kinase	proliferation, angiogenesis
VEGFR2 P35968	oligoMan, Siaα6, O-glycans, binds VEGF	18	Tyr-kinase	proliferation, angiogenesis
VEGFR3 P35916	binds VEGF	12	Tyr-kinase	proliferation, angiogenesis
IGFR1 P08069	biantennary, hybrid, oligoMan, N-glycans, core Fuc, binds insulin, IGF1, IGF2	16	Tyr-kinase	cell growth, survival
INSR P06213	binds insulin, IGF1, IGF2	18	Tyr-kinase	cell growth, survival

Plasma membrane Tyrosine kinase receptors (dimeric): Fuc, fucose; oligoMan, oligomannose-type *N*-glycans; poly-LacNAc, poly-*N*-acetyllactosamine chains; Sia, sialic acid.

**Table 2 cells-10-01252-t002:** Role of glycosylation in EGFR function.

Glycosylation	Cell Type	EGFR Function	Reference
↓ *N*-glycans	cancer	↑ activation	[29]
↓ Asn418	CHO	↑ proliferation	[30]
Mutation at Asn420	A431 epidermoid	↑ EGF-independent proliferation	[32]
Mutation at Asn579	A431 epidermoid	↑ dimerization	[33]
↑ Gn-T III	HeLa	↑ phosphorylation	[34]
↑ Gn-T III	glioma	↓ phosphorylation ↑ proliferation ↓ ligand binding	[35]
↑ Gn-T III	rat pheochromocytoma PC12	↓ neurite outgrowth ↓ activation	[36]
↓ Gn-T V	breast cancer	↓ activation	[37]
↓ FUT8	mouse embryonic fibroblasts	↓ activation ↓ phosphorylation	[38]
↑ FUT8	HEK293	↑ signaling	[39]
↓ FUT1	oral squamous carcinoma	↑ cell migration	[40,41]
↓ FUT1	gastric cancer NCI-N87	↓ cell migration, ↑ degradation, ↓ expression ↓ phosphorylation	[42]
↑ FUT1	ovarian cancer RMG-I	↑ phosphorylation ↑ proliferation	[43]
↓ FUT4	epidermoid cancer A431	↓ phosphorylation ↓ tumor growth	[44]
↓ FUT4	melanoma	↓ phosphorylation ↓ proliferation	[45]
↓ FUT4	bronchial epithelial	↓ phosphorylation	[46]
↑ FUT4, ↑ FUT6	A549 lung cancer	↓ phosphorylation ↓ dimerization	[47]
Fucosidase, sialidase	A549 lung cancer	↑ dimerization ↑ proliferation	[47]
↑ Sialylation	A549 lung cancer	↓ invasion	[47]
↓ FUT1, ↓ FUT4	epidermoid cancer A431	↓ activation ↓ phosphorylation ↓ tumor growth	[44]
↓ Sialylation	lung cancer	↑ phosphorylation ↑ TKI sensitivity	[48]
Sialidase	A549 lung cancer	↑ invasion	[47]
Sialidase	A549 lung cancer	↑ activation	[49,50]
Sialidase	A549 lung cancer	↓ activation	[51]
↑ ST6Gal I	ovarian cancer	↑ activation	[52]
↑ ST6Gal I	pancreatic cancer	↑ activation ↑ EMT	[53]
↓ ST6Gal I	colon cancer	↑ activation	[54]
↑ GALNT2	gastric adenocarcinoma	↓ activation ↓ tumorigenesis	[55]
↓ GALNT2	gastric adenocarcinoma	↑ activation ↑ phosphorylation	[55]
GALNT2	oral squamous cellular carcinoma	migration, invasion	[56]
↓ GALNT2	hepatocellular carcinoma	↓ phosphorylation ↓ activation	[56]
GALNT2	glioma	↑ activation	[57]
↓ GALNT2	glioma	↓ tumor growth ↓ phosphorylation	[57]

EMT, epithelial-mesenchymal transition; Gal, galactose; GALNT, polypeptide GalNAc-transferase; Gn-T, GlcNAc-transferase; FUT, Fuc-transferase; ST6Gal, α6-sialyltransferase.

**Table 3 cells-10-01252-t003:** Neuronal receptors.

Protein Uniprot No.	Extracellular Domain	*N*-glycosylation Sites	Cytoplasmic Domain	Function
RET P07949	Complex N-glycans	12	Tyr-kinase	survival
NTRK1, TrkA P04629	oligoMan, Sia binds NGF	13	Tyr-kinase	proliferation differentiation
NTRK2, TrkB Q16620	binds BDNF, NGF4	11	Tyr-kinase	neuronal development proliferation
NTRK3, TrkC Q16288	binds NGF3	13	Tyr-kinase	survival, differentiation
NGFR, p75 P08138	O-glycans, binds NGF	1	protein-protein interaction	circadian rhythm, apoptosis, differentiation, survival
GFRα1 P56159	binds GDNF	3	GPI anchor	
GFRα2 O00451	binds NRTN binds RET	4	GPI anchor	
GFRα3 O60609	binds RET binds GDNF	4	GPI anchor	
GFRα4 Q9GZZ7	binds RET binds GDNF	1	GPI anchor	

OligoMan, oligomannose-type *N*-glycans; Sia, sialic acid.

**Table 4 cells-10-01252-t004:** Transforming growth factor receptors.

Protein Uniprot No.	Extracellular Domain	*N*-glycosylation Sites	Role	Cytoplasmic Domain
TGFR1 P36897	Tetraantennary N-glycan core Fuc, Fucα1-3 GlcNAcβ1-6 bisected N-glycan binds TGFβ	1	promotes signaling promotes signaling promotes signaling promotes EMT inhibits EMT	Ser/Thr kinase, proliferation binds SMAD2
TGFR II, P37173	oligoMan tetraantennary N-glycans, core Fuc, Fucα1-3, Sia-Lewis^x^, Sia-Lewis^a^ Ser31-O-glycan Binds TGFβ3	3	promotes signaling blocks signaling	Ser/Thr kinase proliferation binds SMAD4
TGFR III Q03167	2 Xyl-Ser GAGs binds TGFβ	5		truncated

GAG, glycosaminoglycans; oligoMan, oligomannose-type *N*-glycans; Sia, sialic acid.

**Table 5 cells-10-01252-t005:** Death receptors.

Protein Uniprot No.	Extracellular Domain	*N*-glycosylation Sites	Cytoplasmic Domain	Function
Fas/CD95 P25445	Siaα6 binds FasL, TNF	2	death domain, DISC binds FADD, caspase-8	apoptosis
TNFRSF25 DR3 Q93038	binds TNF	2	death domain	apoptosis
TNFRSF10A DR4 O00220	O-glycans, Fuc binds TNF, TRAIL	1	death domain, DISC, binds FADD, caspase-8, O-GlcNAc-phosphate at Ser 424	apoptosis
TNFRSF10B DR5 O14763	O-glycans,T, ST antigens, binds TRAIL, TNF	-	death domain, DISC, binds FADD, caspase-8	apoptosis
TNFRSF21 DR6 O75509	O-glycans, binds TRAIL, TNF	6	death domain, DISC, binds FADD, caspase-8	apoptosis
TNFRSF1A TNFR1 P19438	binds TNF	3	death domain, DISC, binds TRADD, FADD	apoptosis, survival
TNFRSF10D Q9UBN6	N-glycans at Asn127,182 binds TRAIL	2	truncated, secreted	-
TNFRSF10C O14798	binds TRAIL	3	truncated, GPI anchored	-

Plasma membrane death receptors of the TNFR superfamily (trimeric): DISC, Death-inducing signaling complex; FADD, Fas-associated protein with death domain; Sia, sialic acid.

## Data Availability

Not applicable.

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
