# Peer review of "Role of Glycans on Key Cell Surface Receptors That Regulate Cell Proliferation and Cell Death"

_cells, 2021, doi:10.3390/cells10051252_

Round 1

Reviewer 1 Report

Authors present a complete, clear and well-written review on the “Role of glycans of cell surface receptors in the regulation of cell proliferation and cell death”.

As written above, the review is complete and well written. Figures and explanatory tables are of great help to understanding. The review deserves publication after some very small corrections:

1) The term High mannoses is no more used. Better to replace it, in the text, in the figure legends and in the tables by the term Oligomannoses.

2) On page 20 and page 22 the authors describe the role of C1GalT, did they want to write C1GalT1?

3) The tables presented are very useful for understanding and schematizing, however in the pdf I downloaded they are not well aligned. I believe that great care must be taken to ensure that they are well aligned and therefore easy to read.

Author Response

Reviewer 1

As written above, the review is complete and well written. Figures and explanatory tables are of great help to understanding. The review deserves publication after some very small corrections:

1) The term High mannoses is no more used. Better to replace it, in the text, in the figure legends and in the tables by the term Oligomannoses.

Thank you for the comments. We changed ‘High mannose’ to ‘Oligomannose’ throughout.

2) On page 20 and page 22 the authors describe the role of C1GalT, did they want to write C1GalT1?

We kept the name ‘C1GalT’ for historic reasons and also because there is only one of these enzymes. COSMC is not a glycosyltransferase although it has been named ‘C1GalT’ before.

3) The tables presented are very useful for understanding and schematizing, however in the pdf I downloaded they are not well aligned. I believe that great care must be taken to ensure that they are well aligned and therefore easy to read.

The Tables were changed when converted to the final layout. We checked all the Tables, and made a number of corrections and hopefully they are well aligned.

We appreciate the reviewer’s comments very much, and realized that it was a long review to work on. We edited the manuscript carefully and corrected many small errors.

We added the keywords. The references were renumbered and in order throughout the Table 2.

Reviewer 2 Report

The review entitled “Role of glycans of cell surface receptors in the regulation of cell proliferation and cell death” focus on the role of glycosylation on ligands binding, affinity and receptors functions in regulating proliferation and cell death. The review is on a timely topic and potentially interesting, due to the lack of recent reviews on the subject. Nevertheless the manuscript needs some revisions and careful formatting. The major concern is the introduction that lacks the fluidity necessary to provide the relevant background information to guide the readers throughout the rest of the manuscript.

  1. The authors claim that this review is focused on role of glycosylation of receptors that regulate proliferation and cell death. With exception of Toll-like receptors, all the remaining receptors included in this review are directly involved either in proliferation or cell death. Regarding the Toll-like receptors, though they are involved in proliferation (1016/j.cyto.2009.08.010), the authors chose to discuss their role in directing immune system to activate anti-microbial immune responses. Since the manuscript is too long, instead of reformulating section 4 towards TLR role in proliferation, I would advise its removal from the manuscript.
  2. The summary needs to be reformulated. The sentences “The review… infection.” Move to the end of summary. In the sentence “The review is examines…” is awkward please change it to something like: “The review aims at examining…” Substitute the word “examine” line 16 by “discuss on”.
  3. The introduction is the weakest part of the manuscript with several loose ideas without a link between the sentences. In lines 37-38 the sentence is out of place. The authors refer to the same subject on lines 49-51. Please reformulate sentence on lines 50-51.
  4. Please provide reference throughout the text and not only by the end of large paragraphs (ex: lines 23-38, 193-200, 565-571, 597-602). In fact there are paragraphs in the introduction without references.
  5. Please reformulate the sentences in lines 97-100 by starting with a more generic sentence such as: “The glycosylation of different families of receptors will be discussed including tyrosine kinase receptors, transforming factor-beta receptors, death receptors and toll-like receptors. Regarding the tyrosine kinase receptors we have selected epidermal growth factor (EGFR), hepatocyte growth factor receptor (MET), … (IGFR).”
  6. In the text (l475) GAGs are mentioned as co-receptors of FGFR and Fig 2A is referred, however no GAGs are represented on Fig 2A. In the same paragraph the authors mention FGFR4 as overexpressed in breast cancer, however no more information is detailed for this receptor isoform in this condition. Alone this sentence makes no sense. Either remove it or elaborate on the subject.
  7. The authors state that the review is focused on plasma membrane receptors in human cells, however there are some references to C. elegans and mouse neuroblastoma cells. Please remove references obtained with receptors from other species.
  8. The authors refer to MCF10A as breast cancer cells (l771). This is not true, in fact MCF10A are used as a model of epithelial breast cells (not carcinoma). In reference 130, to investigate whether FUT8 was functionally involved in regulating the aggressiveness of breast cancer cells, the expression of FUT8 was evaluated in epithelial-like normal human epithelial cells (MCF-10A), low-metastatic breast cancer cells (T-47D) and mesenchymal-like highly invasive breast cancer cell lines (MDA-MB-231 and Hs578T). FUT8 protein level was higher in highly invasive breast cancer cells (MDA-MB-231 and Hs578T) than normal human epithelial cells and low-metastatic breast cancer cells (MCF-10A and T-47D). Please be careful in the statements you use. Please correct the text accordingly.
  9. Figure 2 – missing the signaling cascades of FGFR.
  10. The authors mention in the text (l901) internalization of receptor complex II that is not referred on Fig4. Please include it or remove the reference in the text.
  11. When referring to an abbreviation after the full name, use: full name (abbr), ex. epidermal growth factor (EGFR). Before using the abbreviation Fuc use the full name fucose: ex Fig 1 caption, table 1. Use always the same abbreviation: FUT8 instead of Fut8. C1GALT instead of C1GalT, ST6GAL instead of ST6Gal, COSMC instead of Cosmc, AKT instead of Akt, SMAD2 instead of Smad2

Minor revisions

  1. Down regulation or downregulation? Choose one and use always that in the text.
  2. Line 369-372: the first and third sentences are saying the same. Reference missing in the last sentence.
  3. Line 215: “… with other receptor molecules” instead of another receptor molecule.
  4. Line 475, missing an r in through.
  5. Last paragraph on page 21 and first on page 22 need to be reformulated.
  6. Line 536: correct to “…linked to any of the Asn residues…”
  7. Use at the cell surface and not on the cell surface (l545, l693)
  8. When referring to lectins please use always abbreviations (the authors either use in abbreviation or in full name-557-562, 682, 686, 698).
  9. Line 566 – involves instead of involve.
  10. Line 581 – IGFR1 instead of IFGR1
  11. Lines 590-593 – sentence too long, consider dividing it
  12. Line 596 – remove “of protein”
  13. Line 611 – use “that” instead of “but”
  14. Lines 622-624, please reformulate – ex “…Tyr kinase receptors, while NGFR p75 has no intrinsic Tyr kinase activity…”
  15. Line 624 – include the abbreviation of polysialic acids that is mentioned in line 626.
  16. Line 677 – through instead of involving
  17. Line 684 – instead of “receptor-bound” use “presence of”
  18. Line 689 – please reformulate ex: “…13 N-glycosylation sites, 6 of them (including…”
  19. Lines 722-728 – no references
  20. Lines 778-780 – reformulate the sentence.
  21. Line 798 – remove “resulted”
  22. Lines 806-810 – no reference
  23. Line 826 - “folding and, plasma membrane…”
  24. Line 992-998 – no reference
  25. Please format the manuscript with the same type of letter as there are several paragraphs with different letter style. Please pay attention when dividing the words (when changing line), separate by syllable: lig-and should be li-gand.
  26. Tables should be formatted to align the text in the respective columns. The abbreviations should be included in the end of the table and not the beginning.

Author Response

Reviewer 2

The review entitled “Role of glycans of cell surface receptors in the regulation of cell proliferation and cell death” focus on the role of glycosylation on ligands binding, affinity and receptors functions in regulating proliferation and cell death. The review is on a timely topic and potentially interesting, due to the lack of recent reviews on the subject. Nevertheless the manuscript needs some revisions and careful formatting. The major concern is the introduction that lacks the fluidity necessary to provide the relevant background information to guide the readers throughout the rest of the manuscript.

  1. The authors claim that this review is focused on role of glycosylation of receptors that regulate proliferation and cell death. With exception of Toll-like receptors, all the remaining receptors included in this review are directly involved either in proliferation or cell death. Regarding the Toll-like receptors, though they are involved in proliferation (1016/j.cyto.2009.08.010), the authors chose to discuss their role in directing immune system to activate anti-microbial immune responses. Since the manuscript is too long, instead of reformulating section 4 towards TLR role in proliferation, I would advise its removal from the manuscript.

Thank you very much for the detailed review. We agree with the reviewer. TLRs are a different kind of receptors. Therefore, we deleted the section on TLR, all the relevant references and other statement regarding TLR in the abstract, Introduction and Conclusions. This is making the long review a bit shorter.

We really appreciate the careful review and opportunity to improve the writing.

  1. The summary needs to be reformulated. The sentences “The review… infection.” Move to the end of summary. In the sentence “The review is examines…” is awkward please change it to something like: “The review aims at examining…” Substitute the word “examine” line 16 by “discuss on”.

The summary was rewritten accordingly.

  1. The introduction is the weakest part of the manuscript with several loose ideas without a link between the sentences. In lines 37-38 the sentence is out of place. The authors refer to the same subject on lines 49-51. Please reformulate sentence on lines 50-51.

We agree and deleted / rewrote the Introduction according to the comments.

  1. Please provide reference throughout the text and not only by the end of large paragraphs (ex: lines 23-38, 193-200, 565-571, 597-602). In fact there are paragraphs in the introduction without references.

We added several new references, and also transferred some of the existing references to explain the writing.

  1. Please reformulate the sentences in lines 97-100 by starting with a more generic sentence such as: “The glycosylation of different families of receptors will be discussed including tyrosine kinase receptors, transforming factor-beta receptors, death receptors and toll-like receptors. Regarding the tyrosine kinase receptors we have selected epidermal growth factor (EGFR), hepatocyte growth factor receptor (MET), … (IGFR).”

Corrected.

  1. In the text (l475) GAGs are mentioned as co-receptors of FGFR and Fig 2A is referred, however no GAGs are represented on Fig 2A. In the same paragraph the authors mention FGFR4 as overexpressed in breast cancer, however no more information is detailed for this receptor isoform in this condition. Alone this sentence makes no sense. Either remove it or elaborate on the subject.

Heparin has been added to Fig.2A and the legend.

  1. The authors state that the review is focused on plasma membrane receptors in human cells, however there are some references to C. elegans and mouse neuroblastoma cells. Please remove references obtained with receptors from other species.

The statements on the receptor expression in insect cells and on C. elegans plus reference have been deleted. However, the studies in mice are essential to understand the human receptors and could not be deleted.

  1. The authors refer to MCF10A as breast cancer cells (l771). This is not true, in fact MCF10A are used as a model of epithelial breast cells (not carcinoma). In reference 130, to investigate whether FUT8 was functionally involved in regulating the aggressiveness of breast cancer cells, the expression of FUT8 was evaluated in epithelial-like normal human epithelial cells (MCF-10A), low-metastatic breast cancer cells (T-47D) and mesenchymal-like highly invasive breast cancer cell lines (MDA-MB-231 and Hs578T). FUT8 protein level was higher in highly invasive breast cancer cells (MDA-MB-231 and Hs578T) than normal human epithelial cells and low-metastatic breast cancer cells (MCF-10A and T-47D). Please be careful in the statements you use. Please correct the text accordingly.

Thank you for detecting the mistake. We corrected the wording on normal human epithelial cells MCF10A.

  1. Figure 2 – missing the signaling cascades of FGFR.

The MAPK/ERK signaling was added to FGFR.

  1. The authors mention in the text (l901) internalization of receptor complex II that is not referred on Fig4. Please include it or remove the reference in the text.

We  removed the statement of Fas receptor internalization from the text.

  1. When referring to an abbreviation after the full name, use: full name (abbr), ex. epidermal growth factor (EGFR). Before using the abbreviation Fuc use the full name fucose: ex Fig 1 caption, table 1. Use always the same abbreviation: FUT8 instead of Fut8. C1GALT instead of C1GalT, ST6GAL instead of ST6Gal, COSMC instead of Cosmc, AKT instead of Akt, SMAD2 instead of Smad2

The abbreviations were corrected to use the same throughout.

Minor revisions

  1. Down regulation or downregulation? Choose one and use always that in the text.
  2. Line 369-372: the first and third sentences are saying the same. Reference missing in the last sentence.
  3. Line 215: “… with other receptor molecules” instead of another receptor molecule.
  4. Line 475, missing an r in through.
  5. Last paragraph on page 21 and first on page 22 need to be reformulated.
  6. Line 536: correct to “…linked to any of the Asn residues…”
  7. Use at the cell surface and not on the cell surface (l545, l693)

Thank you, we made all the corrections.

  1. When referring to lectins please use always abbreviations (the authors either use in abbreviation or in full name-557-562, 682, 686, 698).

Many people are not familiar with the abbreviated versions of lectin names.  Reviewer 3 suggested to use the full name and the abbreviations which we did.

  1. Line 566 – involves instead of involve.
  2. Line 581 – IGFR1 instead of IFGR1
  3. Lines 590-593 – sentence too long, consider dividing it
  4. Line 596 – remove “of protein”
  5. Line 611 – use “that” instead of “but”
  6. Lines 622-624, please reformulate – ex “…Tyr kinase receptors, while NGFR p75 has no intrinsic Tyr kinase activity…”
  7. Line 624 – include the abbreviation of polysialic acids that is mentioned in line 626.
  8. Line 677 – through instead of involving
  9. Line 684 – instead of “receptor-bound” use “presence of”
  10. Line 689 – please reformulate ex: “…13 N-glycosylation sites, 6 of them (including…”
  11. Lines 722-728 – no references
  12. Lines 778-780 – reformulate the sentence.
  13. Line 798 – remove “resulted”
  14. Lines 806-810 – no reference
  15. Line 826 - “folding and, plasma membrane…”
  16. Line 992-998 – no reference
  17. Please format the manuscript with the same type of letter as there are several paragraphs with different letter style. Please pay attention when dividing the words (when changing line), separate by syllable: lig-and should be li-gand.

We edited the text according to all of the comments.

  1. Tables should be formatted to align the text in the respective columns. The abbreviations should be included in the end of the table and not the beginning.

The Tables were formated and corrected.

Reviewer 3 Report

The review manuscript by Gao and colleagues, entitled "Role of glycans of cell surface receptors in the regulation of cell proliferation and cell death", gathers a huge number of data on glycosylation of selected cell surface proteins, such as growth factor receptors and death receptors. Authors have done a tremendous work to collect such a large amount of results on the glycosylation of selected membrane receptors. The manuscript is interesting but it was difficult to avoid mistakes, inconsistencies, and repetitions in compiling so many source articles. Manuscript was prepared with less than necessary care. It seems to me that the better idea was to prepare two reviews based on the gathered data and do it more carefully and clearly. The following points should be addressed in the revised manuscript:

(1) The title of the manuscript is too wide, because the term "cell surface receptors" refers also to cell adhesion molecules (mainly integrins and cadherins) which were not included in this review.

(2) Lines 37-38: It is not clear why this sentence “Many receptors have isoforms that differ in amino acid sequence, ligand binding ability and function” appear at the end of the first paragraph.

(3) The names of glycans on Figure 1 are not used consistently; it should be: “Bisected complex-type N-glycan, Tetraantennary complex-type N-glycan, Sialylated biantennary complex-type N-glycan.

(4) Figure caption should not include the data not illustrated on figure, e.g. the sentence “In addition, some receptors have covalent glycosaminoglycan linkages in the extracellular domain as well as O-GlcNAc linked to Ser/Thr residues in the cytoplasmic domain.” (lines 88-90) in the legend to Figure 1 is not visualized on this figure. Figure 1 shows the structure of glycans on cell surface receptors, but not the role of glycans therefore the last sentence in the legend to this Figure is unnecessary. Similarly, some parts of the caption to Figure 2 should be included in the text, not in the legend to this figure, e.g. "The roles of individual structures have been examined by modifications of glycans including inhibition, overexpression and knock down of glycosyltransferases.", and "Cell surface and membrane-bound proteins, glycoproteins and glycolipids participate in the regulation of receptor function.". The legends of Figures 4 and 5 also need removing of sentences which not reflect the schemes on these Figures.

(5) In Tables 1, 3, 4, and 5 I suggest to add to the second column the numbers of N-glycosylated asparagines (e.g. Asn297) instead of a strange abbreviation "N-gly". The data gathered in the second column of these Tables should be divided into two columns named e.g. "receptor glycosylation" and "ligands". Tables (especially Table 2) are wrong organized and it is difficult to assign a function and references to the type of glycosylation. The numbers below receptor abbreviations in Tables 1, 3,4, and 5 are probably Uniprot numbers, like in Table 6. If yes, this information should be added to the name of columns.

(6) The legend to glycan symbols should be added to Figure 2. I guess that on EGFR two N-glycans are shown, one of them is a high-mannose structure, the second one is a tetraantennary complex-type N-glycan. While in the Figure caption is written that "Only one of the multiple N-glycans on EGFR is shown." (lines: 179-180).

(7) It is not clear the meaning of MET abbreviation. It is used both as the abbreviation of mesenchymal-epithelial transition (line 419) and the abbreviation of hepatocyte growth factor receptor (e.g. line 418).

(8) The spelling of the names of glycans, bonds, and structures requires standardization throughout the manuscript.

(9) A lack of references in some parts of the manuscript, e.g. in the first paragraph of 1d section.

(10) The abbreviations should be carefully checked, e.g.:

- the full names of monosaccharides (GlcNAc, Gal, GalNAc and Fuc) was not provided,

- in turn thee full name of other abbreviations were provided more than one time, e.g. GT/GTs (lines 47 and 73), GPI (lines 101and 671),

- the commonly used abbreviation of poly-N-acetyllactosamine chains is poly-LacNAc but not polylac (line 122), then it should be used in the rest of the manuscript, e.g. lines 505-509,

- the abbreviation of E-phytohemagglutinin, Phaseolus vulgaris lectin (PHA-E) (line 682) should be provided and the full name of PHA-L (line 686) should be added,

- the full name of EMT was not provided.

(11) The manuscript should be also checked carefully for small mistakes such and inconsistency e.g.:

- lines 98-107: the abbreviation of growth factor receptors should be provided in parentheses,

- uppercase and lowercase letters should be corrected throughout the manuscript, e.g. it should be fucose not Fucose (line 90), COSMIC (e.g. line 551) or Cosmic (e.g. line 927), and many others,

- lines 121-122: the expression "potential N-glycosylation sites" is better instead of "N-glycosylation sites that may be glycosylated",

- line 465: in O-glycan core 1 structure "beta" was missed, it should be: Galbeta1-3GalNAc,

- line 925: type of linkage should be added to T antigen: Galbeta1,3GalNAc

- the abbreviations of gene names should be italicized, e.g. lines 389 and 542: ST6Gal I, line 438: Fut8, line 441: GnT-III, line 515: C1GalT, and many others,

- Latin names (e.g. lines 455 and 558: Vicia villosa) should be italicized,

- typos and punctuation errors are quite common.

Author Response

Reviewer 3:

The review manuscript by Gao and colleagues, entitled "Role of glycans of cell surface receptors in the regulation of cell proliferation and cell death", gathers a huge number of data on glycosylation of selected cell surface proteins, such as growth factor receptors and death receptors. Authors have done a tremendous work to collect such a large amount of results on the glycosylation of selected membrane receptors. The manuscript is interesting but it was difficult to avoid mistakes, inconsistencies, and repetitions in compiling so many source articles. Manuscript was prepared with less than necessary care. It seems to me that the better idea was to prepare two reviews based on the gathered data and do it more carefully and clearly. The following points should be addressed in the revised manuscript:

The section on TLR was removed, making the text shorter. We apologize for the many little mistakes and have corrected the mistakes. inconsistencies and redundancies.

(1) The title of the manuscript is too wide, because the term "cell surface receptors" refers also to cell adhesion molecules (mainly integrins and cadherins) which were not included in this review.

We changed the title to ‘Role of glycans of cell surface receptors that regulate cell proliferation and cell death’

(2) Lines37-38: It is not clear why this sentence “Many receptors have isoforms that differ in amino acid sequence, ligand binding ability and function” appear at the end of the first paragraph.

The Introduction was rewritten and the sentence was removed.

(3) The names of glycans on Figure 1 are not used consistently; it should be: “Bisected complex-type N-glycan, Tetraantennary complex-type N-glycan, Sialylated biantennary complex-type N-glycan.

We corrected the glycan names.

(4) Figure caption should not include the data not illustrated on figure, e.g. the sentence “In addition, some receptors have covalent glycosaminoglycan linkages in the extracellular domain as well as O-GlcNAc linked to Ser/Thr residues in the cytoplasmic domain.” (lines 88-90) in the legend to Figure 1 is not visualized on this figure. Figure 1 shows the structure of glycans on cell surface receptors, but not the role of glycans therefore the last sentence in the legend to this Figure is unnecessary. Similarly, some parts of the caption to Figure 2 should be included in the text, not in the legend to this figure, e.g. "The roles of individual structures have been examined by modifications of glycans including inhibition, overexpression and knock down of glycosyltransferases.", and "Cell surface and membrane-bound proteins, glycoproteins and glycolipids participate in the regulation of receptor function.". The legends of Figures 4 and 5 also need removing of sentences which not reflect the schemes on these Figures.

Thank you for the comments. We edited all the figure legends accordingly.

(5) In Tables 1, 3, 4, and 5 I suggest to add to the second column the numbers of N-glycosylated asparagines (e.g. Asn297) instead of a strange abbreviation "N-gly". The data gathered in the second column of these Tables should be divided into two columns named e.g. "receptor glycosylation" and "ligands". Tables (especially Table 2) are wrong organized and it is difficult to assign a function and references to the type of glycosylation. The numbers below receptor abbreviations in Tables 1, 3,4, and 5 are probably Uniprot numbers, like in Table 6. If yes, this information should be added to the name of columns.

We added a column on the number of N-glycosylation sites to Table 2, and for all Tables explained the Uniprot No. However we left all the features of the extracellular domain in one column, as the Table would be too crowded with columns otherwise.

(6) The legend to glycan symbols should be added to Figure 2. I guess that on EGFR two N-glycans are shown, one of them is a high-mannose structure, the second one is a tetraantennary complex-type N-glycan. While in the Figure caption is written that "Only one of the multiple N-glycans on EGFR is shown." (lines: 179-180).

The glycan symbols were added. We changed the figure legend accordingly and deleted ‘multiple’.

(7) It is not clear the meaning of MET abbreviation. It is used both as the abbreviation of mesenchymal-epithelial transition (line 419) and the abbreviation of hepatocyte growth factor receptor (e.g. line 418).

We redefined MET as the hepatocyte growth factor receptor.

(8) The spelling of the names of glycans, bonds, and structures requires standardization throughout the manuscript.

(9) A lack of references in some parts of the manuscript, e.g. in the first paragraph of 1d section.

(10) The abbreviations should be carefully checked, e.g.:

- the full names of monosaccharides (GlcNAc, Gal, GalNAc and Fuc) was not provided,

- in turn thee full name of other abbreviations were provided more than one time, e.g. GT/GTs (lines 47 and 73), GPI (lines 101and 671),

- the commonly used abbreviation of poly-N-acetyllactosamine chains is poly-LacNAc but not polylac (line 122), then it should be used in the rest of the manuscript, e.g. lines 505-509,

- the abbreviation of E-phytohemagglutinin, Phaseolus vulgaris lectin (PHA-E) (line 682) should be provided and the full name of PHA-L (line 686) should be added,

- the full name of EMT was not provided.

(11) The manuscript should be also checked carefully for small mistakes such and inconsistency e.g.:

- lines 98-107: the abbreviation of growth factor receptors should be provided in parentheses,

- uppercase and lowercase letters should be corrected throughout the manuscript, e.g. it should be fucose not Fucose (line 90), COSMIC (e.g. line 551) or Cosmic (e.g. line 927), and many others,

- lines 121-122: the expression "potential N-glycosylation sites" is better instead of "N-glycosylation sites that may be glycosylated",

- line 465: in O-glycan core 1 structure "beta" was missed, it should be: Galbeta1-3GalNAc,

- line 925: type of linkage should be added to T antigen: Galbeta1,3GalNAc

- the abbreviations of gene names should be italicized, e.g. lines 389 and 542: ST6Gal I, line 438: Fut8, line 441: GnT-III, line 515: C1GalT, and many others,

- Latin names (e.g. lines 455 and 558: Vicia villosa) should be italicized,

- typos and punctuation errors are quite common.

Thank you very much for the careful review. We made the corrections to the manuscript according to all of these comments.

Round 2

Reviewer 3 Report

The majority of my suggestions were included in the revised manuscript by Gao and colleagues and the manuscript is close to be accepted for publication. The title of the manuscript was changed but in my opinion it does still not exclude e.g. integrins which are known to be responsible for cell proliferation and death as it was presented in many previous papers. English is not my first language but it seems to me that we can say that integrins are also "cell surface receptors that regulate cell proliferation and cell death". I suggest to add to the title e.g. main/primary/pivotal/selected cell surface receptors.

(1) I also suggest to:

- italicize "N-" and "O-" in the nomenclature of glycans (types of glycans and glycosylation, types of glycosidic bonds) as it is it increasingly used,

- use the forms with dash "complex-type" and hybrid-type", not "complex type" and hybrid type",

- add the full names for all abbreviations used in a figure in the caption to this figure.

(2) The text still needs careful checking, e.g. it should be:

- lines 444-445: sialyl-Lewisx, not Sialyl-Lewisx,

- line 507 and 930: sialyl-Tn, not Sialyl-Tn,

- line: 364: sialyl-Lewisx, not sialyl- Lewisx, etc.

Author Response

Thank you very much for the additional comments.

We changed the title to: Role of glycans of key cell surface receptors that regulate cell proliferation and cell death

We changed figure 2A, C to write AKT and to include HS in the legend.

We italicized the N- and O-for all glycans.

We added the full names of abbreviations in the figure legends.

Complex-type and Hybrid-type were edited to include the hyphen.

We corrected the capital S in sialyl- and corrected the spacing.